# A novel computational pipeline for *var* gene expression augments the discovery of changes in the *Plasmodium falciparum* transcriptome during transition from in vivo to short-term in vitro culture

Clare Andradi-Brown[1,2,3], Jan Stephan Wichers-Misterek[4,5,6], Heidrun von Thien[4,5,6], Yannick D Höppner[4,5,6], Judith AM Scholz[4], Helle Hansson[7,8], Emma Filtenborg Hocke[7,8], Tim Wolf Gilberger[4,5,6], Michael F Duffy[9], Thomas Lavstsen[7,8], Jake Baum[2,10], Thomas D Otto[11†], Aubrey J Cunnington[1,3*†], Anna Bachmann[4,5,6,12*†]

[1]Section of Paediatric Infectious Disease, Department of Infectious Disease, Imperial College London, London, United Kingdom; [2]Department of Life Sciences, Imperial College London, South Kensington, London, United Kingdom; [3]Centre for Paediatrics and Child Health, Imperial College London, London, United Kingdom; [4]Bernhard Nocht Institute for Tropical Medicine, Bernhard-Nocht-Strasse, Hamburg, Germany; [5]Centre for Structural Systems Biology, Hamburg, Germany; [6]Biology Department, University of Hamburg, Hamburg, Germany; [7]Center for Medical Parasitology, Department of Immunology and Microbiology, University of Copenhagen, Copenhagen, Denmark; [8]Department of Infectious Diseases, Copenhagen University Hospital, Copenhagen, Denmark; [9]Department of Microbiology and Immunology, University of Melbourne, Melbourne, Australia; [10]School of Biomedical Sciences, Faculty of Medicine & Health, UNSW, Kensington, Sydney, United Kingdom; [11]School of Infection & Immunity, MVLS, University of Glasgow, Glasgow, United Kingdom; [12]German Center for Infection Research (DZIF), partner site Hamburg-Borstel-Lübeck-Riems, Hamburg, Germany

*For correspondence:
a.cunnington@imperial.ac.uk
(AJC);
bachmann@bni-hamburg.de (AB)

†These authors contributed
equally to this work

Competing interest: The authors
declare that no competing
interests exist.

Reviewing Editor: Urszula
Krzych, Walter Reed Army
Institute of Research, United
States

**Abstract** The pathogenesis of severe *Plasmodium falciparum* malaria involves cytoadhesive microvascular sequestration of infected erythrocytes, mediated by *P. falciparum* erythrocyte membrane protein 1 (PfEMP1). PfEMP1 variants are encoded by the highly polymorphic family of *var* genes, the sequences of which are largely unknown in clinical samples. Previously, we published new approaches for *var* gene profiling and classification of predicted binding phenotypes in clinical *P. falciparum* isolates (Wichers et al., 2021), which represented a major technical advance. Building on this, we report here a novel method for *var* gene assembly and multidimensional quantification from RNA-sequencing that outperforms the earlier approach of Wichers et al., 2021, on both laboratory and clinical isolates across a combination of metrics. Importantly, the tool can interrogate the *var* transcriptome in context with the rest of the transcriptome and can be applied to enhance our understanding of the role of *var* genes in malaria pathogenesis. We applied this new method to investigate changes in *var* gene expression through early transition of parasite isolates to in vitro culture, using paired sets of ex vivo samples from our previous study, cultured for up to three generations. In parallel, changes in non-polymorphic core gene expression were investigated. Modest but unpredictable *var* gene switching and convergence towards *var2csa* were observed in culture, along

with differential expression of 19% of the core transcriptome between paired ex vivo and generation 1 samples. Our results cast doubt on the validity of the common practice of using short-term cultured parasites to make inferences about in vivo phenotype and behaviour.

## eLife assessment

Focusing mainly on var genes, the investigators performed comprehensive computational analyses of gene expression in malaria parasites isolated from patients and assessed changes that occur as these parasites adapt to in vitro culture conditions. The study provides an improved computational pipeline for monitoring var gene expression, and importantly, the study documents changes in expression of the core genome and thus provides **important** insights into metabolic adaptations that parasites undergo while transitioning to culture conditions. The findings are **important** for their technical advances that are more rigorous than the current state-of-the-art. The **solid** data analyses, broadly support the claims with only minor weaknesses, tell us to be cautious when interpreting results obtained only from cultured parasites.

## Introduction

Malaria is a parasitic life-threatening disease caused by species of the *Plasmodium* genus. In 2021, there were an estimated 619,000 deaths due to malaria, with children under 5 accounting for 77% of these (*WHO, 2022*). *Plasmodium falciparum* causes the greatest disease burden and most severe outcomes, but our efforts to combat the disease are challenged by its complex life cycle and its sophisticated immune evasion strategies. *P. falciparum* has several highly polymorphic variant surface antigens encoded by multi-gene families, with the best studied high molecular weight *P. falciparum* erythrocyte membrane protein 1 (PfEMP1) family of proteins known to play a major role in the pathogenesis of malaria (*Leech et al., 1984*; *Wahlgren et al., 2017*). About 60 polymorphic *var* genes per parasite genome encode different PfEMP1 variants, which are exported to the surface of parasite-infected erythrocytes, where they mediate cytoadherence to host endothelial cells (*Leech et al., 1984*; *Su et al., 1995*; *Smith et al., 1995*; *Baruch et al., 1995*; *Rask et al., 2010*). *Var* genes are expressed in a mutually exclusive pattern, resulting in each parasite expressing only one *var* gene, and therefore one PfEMP1 protein, at a time (*Scherf et al., 1998*). Due to the exposure of PfEMP1 proteins to the host immune system, switching expression between the approximately 60 *var* genes in the genome is an effective immune evasion strategy, which can result in selection and dominance of parasites expressing particular *var* genes within each host (*Smith et al., 1995*).

Despite their sequence polymorphism, *var* genes could be classified into four categories (A, B, C, and E) according to their chromosomal location, transcriptional direction, type of 5'-upstream sequence (UPSA–E), and encoded protein domains with associated binding phenotype (*Figure 1*; *Lavstsen et al., 2003*; *Kraemer and Smith, 2003*; *Kyes et al., 2007*; *Rask et al., 2010*). PfEMP1 proteins have up to 10 extracellular domains, with the N-terminal domains forming a semi-conserved head structure complex typically containing the N-terminal segment (NTS), a Duffy binding-like domain of class α (DBLα) coupled to a cysteine-rich interdomain region (CIDR). C-terminally to this head structure, PfEMP1 proteins exhibit a varying but semi-ordered composition of additional DBL and CIDR domains of different subtypes (*Figure 1c*). The PfEMP1 family divides into three main groups based on the receptor specificity of the N-terminal CIDR domain: (i) PfEMP1 proteins with CIDRα1 domains bind endothelial protein C receptor (EPCR), while (ii) PfEMP1 proteins with CIDRα2–6 domains bind CD36 and (iii) the atypical VAR2CSA PfEMP1 proteins bind placental chondroitin sulphate A (CSA) (*Salanti et al., 2004*). In addition to these, a subset of PfEMP1 proteins have N-terminal CIDRβ/γ/δ domains of unknown function. This functional diversification correlates with the genetic organisation of the *var* genes. Thus, UPSA *var* genes encode PfEMP1 proteins with domain sub-variants NTSA-DBLα1-CIDRα1/β/γ/δ, whereas UPSB and UPSC *var* genes encode PfEMP1 proteins with NTSB-DBLα0-CIDRα2–6. One exception to this rule is the B/A chimeric *var* genes, which encode NTSB-DBLα2-CIDRα1 domains. The different receptor binding specificities are associated with different clinical outcomes of infection. Pregnancy-associated malaria is linked to parasites expressing VAR2CSA, whereas parasites expressing EPCR-binding PfEMP1 are linked to severe malaria and parasites expressing CD36-binding PfEMP1 are linked to uncomplicated malaria (*Turner et al., 2013*; *Lavstsen et al., 2012*; *Avril et al.,*

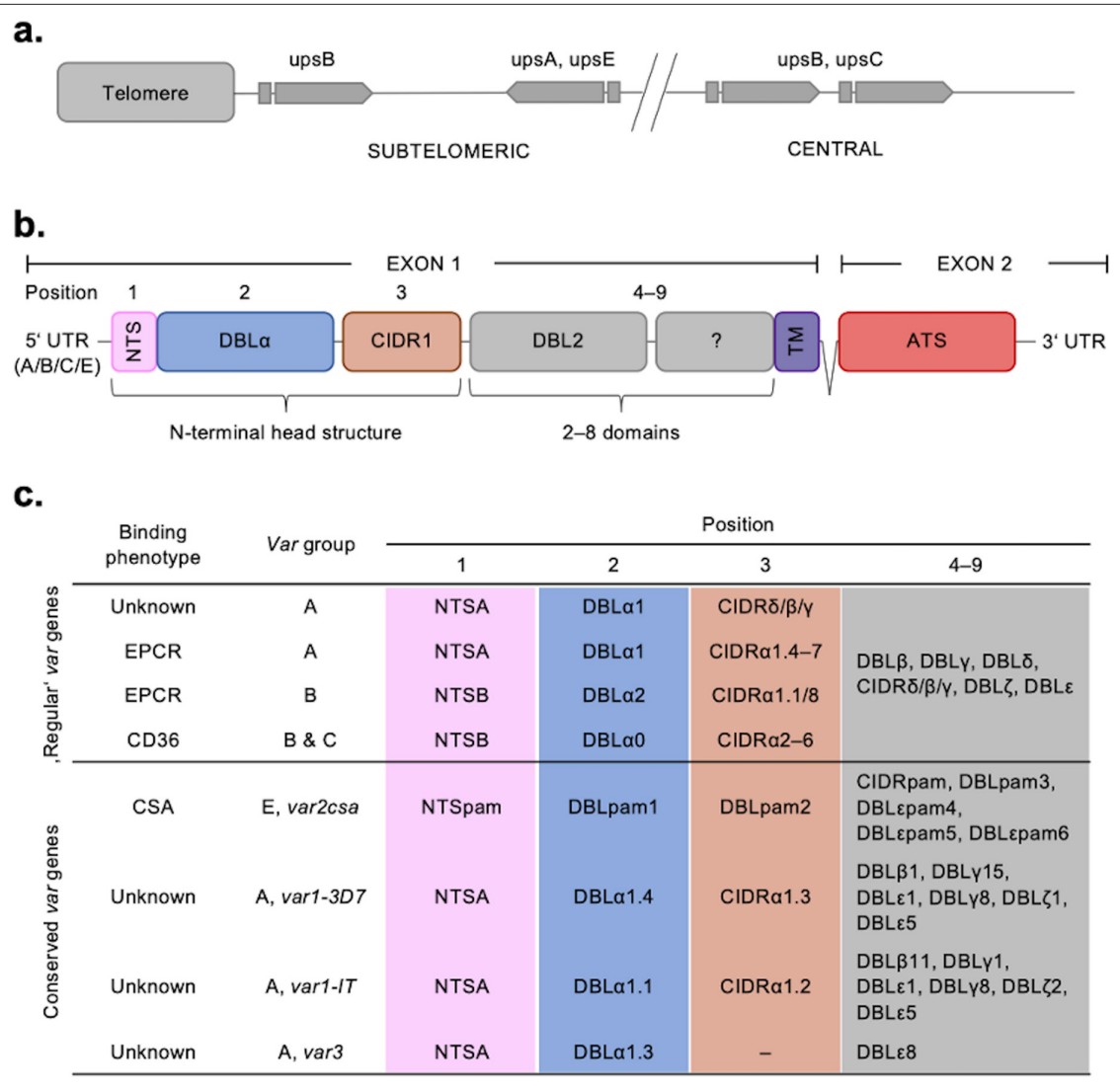

**Figure 1.** Summary of the *var* chromosomal location, *var* gene, PfEMP1 protein structure, and PfEMP1-binding phenotypes. (**a**) Chromosomal position and transcriptional direction (indicated by arrows) of the different *var* gene groups, designated by the respective type of upstream sequence (**Kraemer and Smith, 2003**; **Lavstsen et al., 2003**). (**b**) Structure of the *var* gene which encodes the PfEMP1 protein. The *var* gene is composed of two exons, the first, around 3–9.4 kb, encodes the highly variable extracellular region and the transmembrane region (TM) of PfEMP1. Exon 2 is shorter with about 1.2 kb and encodes a semi-conserved intracellular region (acidic terminal segment [ATS]). The PfEMP1 protein is composed of an N-terminal segment (NTS), followed by a variable number of Duffy binding-like (DBL) domains and cysteine-rich interdomain regions (CIDR) (**Rask et al., 2010**). (**c**) Summary of PfEMP1 proteins encoded in the parasite genome, their composition of domain subtypes, and associated N-terminal binding phenotype. Group A and some B proteins have an EPCR-binding phenotype; the vast majority of group B and C PfEMP1 proteins bind to CD36. Group A proteins also include those that bind a yet unknown receptor, as well as VAR1 and VAR3 variants with unknown function and binding phenotype. VAR2CSA (group E) binds placental chondroitin sulphate A (CSA).

2012; **Claessens et al., 2012**; **Tonkin-Hill et al., 2018**; **Wichers et al., 2021**). The clinical relevance of PfEMP1 proteins with unknown binding phenotypes of the N-terminal head structure and C-terminal PfEMP1 domains is largely unknown, albeit specific interactions with endothelial receptors and plasma proteins have been described (**Tuikue Ndam et al., 2017**; **Quintana et al., 2019**; **Stevenson et al., 2015**). Each parasite genome carries a similar repertoire of *var* genes, which in addition to the described variants include a highly conserved *var1* variant of either type 3D7 or IT, which in most genomes occurs with a truncated or absent exon 2. Also, most genomes carry the unusually small and highly conserved *var3* genes, of unknown function (**Figure 1c**; **Otto et al., 2019**).

Comprehensive characterisation and quantification of *var* gene expression in field samples have been complicated by biological and technical challenges. The extreme polymorphism of *var* genes precludes a reference *var* sequence. *Var* genes can be lowly expressed or not expressed at all, contain repetitive domains, and can have large duplications (*Otto et al., 2019*). Consequently, most studies relating *var* gene expression to severe malaria have relied on primers with restricted coverage of the *var* family, use of laboratory-adapted parasite strains, or have predicted the downstream sequence from DBLα domains (*Sahu et al., 2021*; *Storm et al., 2019*; *Shabani et al., 2017*; *Mkumbaye et al., 2017*; *Kessler et al., 2017*; *Bernabeu et al., 2016*; *Jespersen et al., 2016*; *Lavstsen et al., 2012*). This has resulted in incomplete *var* gene expression quantification and the inability to elucidate specific or detect atypical *var* sequences. RNA-sequencing has the potential to overcome these limitations and provide a better link between *var* expression and PfEMP1 phenotype in in vitro assays, co-expression with other genes or gene families and epigenetics. While approaches for *var* assembly and quantification based on RNA-sequencing have recently been proposed (*Wichers et al., 2021*; *Stucke et al., 2021*; *Andrade et al., 2020*; *Tonkin-Hill et al., 2018*; *Duffy et al., 2016*), these still produce inadequate assembly of the biologically important N-terminal domain region, have a relatively high number of misassemblies, and do not provide an adequate solution for handling the conserved *var* variants (*Table 1*).

*Plasmodium* parasites from human blood samples are often adapted to or expanded through in vitro culture to provide sufficient parasites for subsequent investigation of parasite biology and phenotype (*Brown and Guler, 2020*). This is also the case for several studies assessing the PfEMP1 phenotype of parasites isolated from malaria-infected donors (*Pickford et al., 2021*; *Joste et al., 2020*; *Storm et al., 2019*; *Tuikue Ndam et al., 2017*; *Bruske et al., 2016*; *Claessens et al., 2012*; *Lavstsen et al., 2005*; *Jensen et al., 2004*; *Kirchgatter and Hernando del, 2002*; *Dimonte et al., 2016*; *Hoo et al., 2019*). However, in vitro conditions are considerably different to those found in vivo, e.g., in terms of different nutrient availability and lack of a host immune response (*Brown and Guler, 2020*). Previous studies found inconsistent results in terms of whether *var* gene expression is impacted by culture and, if so, which *var* groups were the most affected (*Zhang et al., 2011*; *Peters et al., 2007*). Similar challenges apply to the understanding of changes in *P. falciparum* non-polymorphic core genes in culture, with the focus previously being on long-term laboratory-adapted parasites (*Claessens et al., 2017*; *Mackinnon et al., 2009*). Consequently, direct interpretation of a short-term cultured parasite's transcriptome remains a challenge. It is fundamental to understand *var* genes in context with the parasite's core transcriptome. This could provide insights into *var* gene regulation and phenomena such as the proposed lower level of *var* gene expression in asymptomatic individuals (*Almelli et al., 2014*; *Andrade et al., 2020*).

Here, we present an improved method for assembly, characterisation, and quantification of *var* gene expression from RNA-sequencing data. This new approach overcomes previous limitations and outperforms current methods, enabling a much greater understanding of the *var* transcriptome. We demonstrate the power of this new approach by evaluating changes in *var* gene expression of paired samples from clinical isolates *of P. falciparum* during their early transition to in vitro culture, across several generations. The use of paired samples, which are genetically identical and hence have the same *var* gene repertoire, allows validation of assembled transcripts and direct comparisons of expression. We complement this with a comparison of changes which occur in the non-polymorphic core transcriptome over the same transition into culture. We find a background of modest changes in *var* gene expression with unpredictable patterns of *var* gene switching, favouring an apparent convergence towards *var2csa* expression. More extensive changes were observed in the core transcriptome during the first cycle of culture, suggestive of a parasite stress response.

## Results

To extend our ability to characterise *var* gene expression profiles and changes over time in clinical *P. falciparum* isolates, we set out to improve current assembly methods. Previous methods for assembling *var* transcripts have focussed on assembling whole transcripts (*Tonkin-Hill et al., 2018*; *Wichers et al., 2021*; *Guillochon et al., 2022*; *Andrade et al., 2020*). However, due to the diversity within PfEMP1 domains, their associations with disease severity, and the fact different domain types are not inherited together, a method focussing on domain assembly was first developed. In addition, a novel whole transcript approach, using a different de novo assembler, was developed and their performance

**Table 1.** Comparison of previous var assembly approaches based on DNA- and RNA-sequencing.

| Study | Assembler | k-mer | Transcript or gene assembly | Validation on reference strain(s) (Yes/No) | Validation on field strain(s) (Yes/No) | Validation across different expression levels (Yes/No) | Read length (Short/Long) | Read correction (Yes/No) | Scaffolding (Yes/No) | Var transcript filter approach | Assumption | Other limitations |
|---|---|---|---|---|---|---|---|---|---|---|---|---|
| Duffy et al., 2016 | Oases | | Transcript | No | No | No | Short | Yes | Yes | Aligned to 399 var genes with BLAST (e-value <$10^{-5}$) | | – No quantification of misassemblies; – Unable to recover full-length transcript assemblies |
| Dara et al., 2017 | Sprai and Celera (no longer maintained) | 71 | Gene | Yes (strain NF54) | Yes (12 UM patient samples) | NA – only genome assemblies | Long and short | Method assumes combination of long- and short-read sequencing will identify errors | No | >500 bp and aligned to VarDom database | Assumes a whole genome assembly is available | – Require prior filtering of human DNA; – Need a combination of short-read and long-read sequencing |
| Tonkin-Hill et al., 2018* | SoapDeNovo-Trans | 21, 31, 41, 52, and 61 | Transcript | Yes (strain ITG) | No | No | Short | No | No | >500 bp and containing a sig. annotated var domain | | – Unable to fully resolve N-terminus; – No quantification of misassemblies |
| Otto et al., 2019 | Masurca +post-assembly improvements | Default | Gene | Yes (clone 3D7) | Yes (15 Pf3k reference genomes) | No | Short | Yes | Yes | | Whole genome dataset | |
| Andrade et al., 2020 | Velvet | 41 | Transcript | No | No | No | Short | No | No | Aligned to VarDom database | | – No quantification of misassemblies |
| Stucke et al., 2021 | rnaSPAdes | Default | Transcript | No | Yes (6 UM patient samples) | No – only the most expressed var gene | Short | Yes | Unclear | >500 bp and containing a sig. annotated var domain | Information about the true var annotation is available | – Performs de novo assembly on all non-human and P. falciparum mapping reads; – Inconsistent results in three samples when comparing genomic and RNA-sequencing results for dominant var gene |

*Also used in **Wichers et al., 2021; Guillochon et al., 2022.**
UM: uncomplicated malaria.

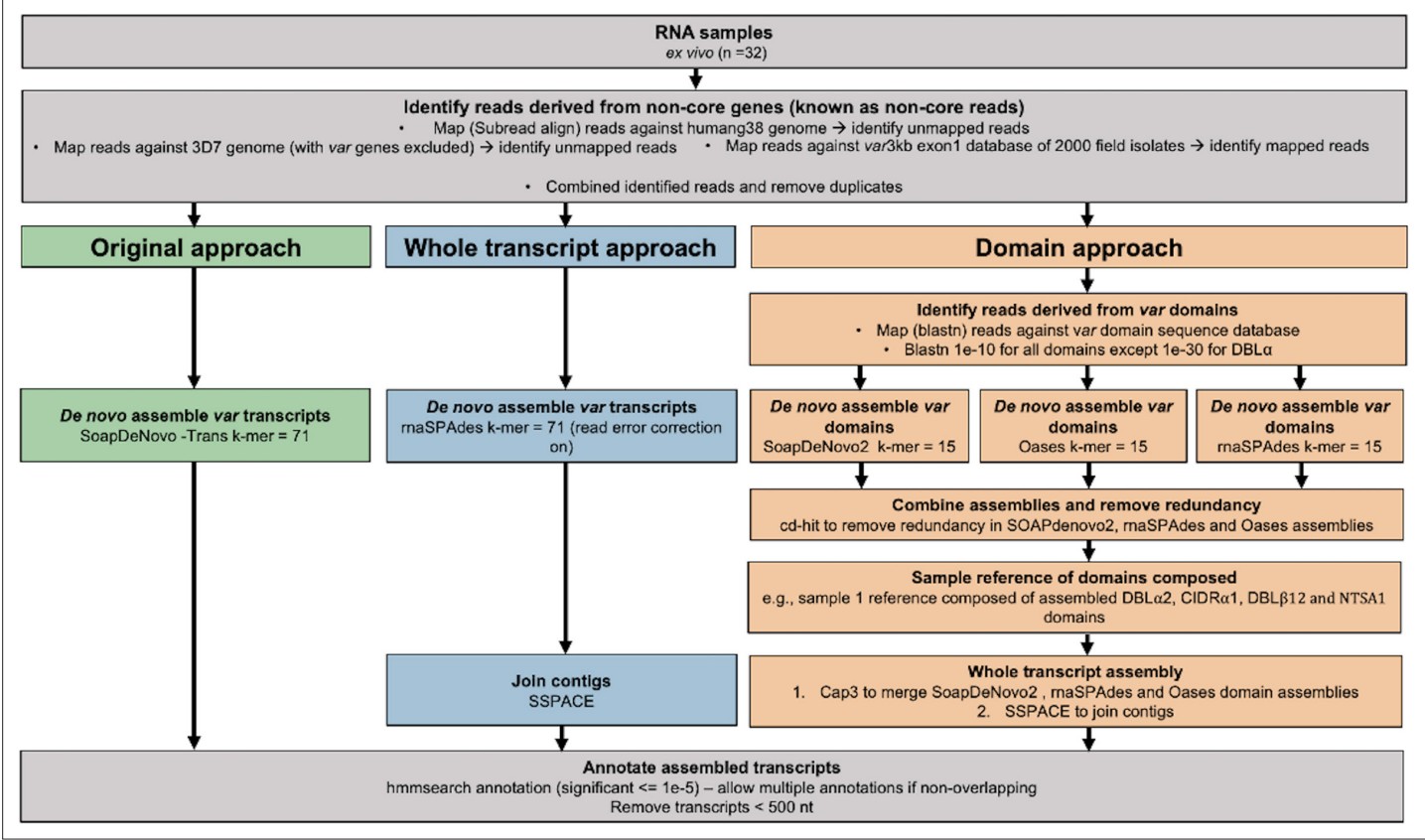

**Figure 2.** Overview of novel computational pipelines for assembling *var* transcripts. The original approach (green) used SoapDeNovo-Trans (k=71) to perform whole *var* transcript assembly. The whole transcript approach (blue) focussed on assembling whole *var* transcripts from the non-core reads using rnaSPAdes (k=71). Contigs were then joined into longer transcripts using SSPACE. The domain approach (orange) assembled *var* domains first and then joined the domains into whole transcripts. Domains were assembled separately using three different de novo assemblers (SoapDeNovo2, Oases, and rnaSPAdes). Next, a reference of assembled domains was composed and cd-hit (at sequence identity = 99%) was used to remove redundant sequences. Cap3 was used to merge and extend domain assemblies. Finally, SSPACE was used to join domains together. Hidden Markov models (HMM) built on the **Rask et al., 2010**, dataset were used to annotate the assembled transcripts (**Rask et al., 2010**). The most significant alignment was taken as the best annotation for each region of the assembled transcript (significance ≤ 1e-5) identified using cath-resolve-hits0. Transcripts <500 nt were removed. A *var* transcript was selected if it contained at least one significantly annotated domain (in exon 1). *Var* transcripts that encoded only the more conserved exon 2 (acidic terminal segment [ATS] domain) were discarded. The three pipelines were run on the 32 malaria patient ex vivo samples from **Wichers et al., 2021**.

The online version of this article includes the following figure supplement(s) for figure 2:

**Figure supplement 1.** Performance of novel computational pipelines for *var* assembly on *P. falciparum* 3D7.

**Figure supplement 2.** Histograms of the length and frequency of *var* transcripts produced by the different *var* assembly approaches.

**Figure supplement 3.** Domain expression correlation between the whole transcript approach and the original approach.

**Figure supplement 4.** *Var* transcript expression differences derived from the whole transcript approach between (**a**) pre-exposed (n=17) (blue) vs naïve (n=15) (orange) cases and (**b**) severe (n=8) (orange) vs non-severe (n=24) (blue) cases.

compared to the method of Wichers et al. (hereafter termed 'original approach', *Figure 2*; *Wichers et al., 2021*). The new approaches made use of the MalariaGEN *P. falciparum* dataset, which led to the identification of additional multi-mapping non-core reads (a median of 3955 reads per sample) prior to *var* transcript assembly (*Ahouidi et al., 2021*). We incorporated read error correction and improved large scaffold construction with fewer misassemblies (see Materials and methods). We then applied this pipeline to paired ex vivo and short-term in vitro cultured parasites to enhance our understanding of the impact of short-term culturing on the *var* transcriptome (*Figure 3*). The *var* transcriptome was assessed at several complementary levels: first, changes in the dominantly expressed *var* gene and the homogeneity of the *var* expression profile in paired samples were investigated; second, changes in *var* domain expression through culture were assessed; and third, *var* group and global *var* gene

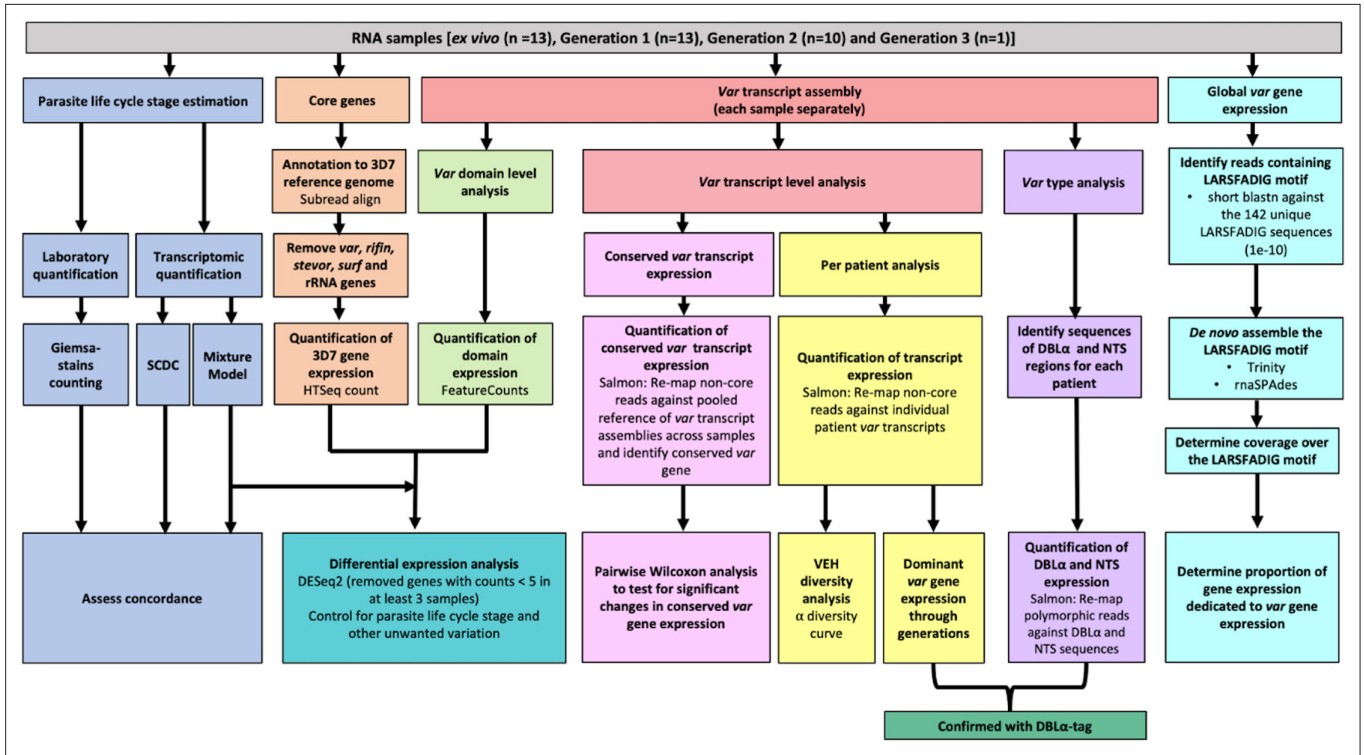

**Figure 3.** Summary of analyses of *var* and core gene transcriptome changes in paired ex vivo and short-term in vitro cultured parasites. From a total of 13 parasite isolates, the ex vivo samples (**Wichers et al., 2021**) and the corresponding in vitro cultured parasites of the first (n=13), second (n=10), and third (n=1) replication cycle were analysed by RNA-sequencing. The expression of non-polymorphic core genes and polymorphic *var* genes was determined in different analysis streams: (1) non-polymorphic core gene reads were mapped to the 3D7 reference genome, expression was quantified using HTSeq and differential expression analysis performed (orange); (2) non-core reads were identified, whole transcripts were assembled with rnaSPAdes, expression of both *var* transcripts (red) and domains (light green) was quantified, and *var* domain differential expression analysis was performed. 'Per patient analysis' (yellow) represents combining all assembled *var* transcripts for samples originating from the same ex vivo sample only. For each conserved *var* gene (var1-3D7, var1-IT, var2csa, and var3), all significantly assembled conserved *var* transcripts were identified and put into a combined reference (pink). The normalised counts for each conserved gene were summed. Non-core reads were mapped to this and DESeq2 normalisation performed. *Var* type (group A vs groups B and C) expression (purple) was quantified using the Duffy binding-like domain of class α (DBLα) and N-terminal segment (NTS) assembled sequences and differences across generations were assessed. Total *var* gene expression (turquoise) was quantified by assembling and quantifying the coverage over the highly conserved LARSFADIG motif, with the performance of assembly using Trinity and rnaSPAdes assessed. DBLα-tag data was used to confirm the results of the dominant *var* gene expression analysis and the *var* type analysis (dark green). *Var* expression homogeneity (VEH) was analysed at the patient level (α diversity curves). All differential expression analyses were performed using DESeq2.To ensure a fair comparison of samples, which may contain different proportions of life cycle stages, the performance of two different in silico approaches was evaluated by counting Giemsa-stained thin blood smears (blue).

The online version of this article includes the following figure supplement(s) for figure 3:

**Figure supplement 1.** Parasite life cycle stage estimates across the generations.

**Figure supplement 2.** *Var* expression profiles across different mapping approaches.

expression changes were evaluated. All these analyses on *var* expression were accompanied by analysis of the core transcriptome at the transition to short-term culture.

## Improving *var* transcript assembly, annotation, and quantification

A laboratory and a clinical dataset were used to assess the performance of the different *var* assembly pipelines (**Figure 2**). The laboratory dataset was a *P. falciparum* 3D7 time course RNA-sequencing dataset (European nucleotide archive [ENA]: PRJEB31535) (**Wichers et al., 2019**). The clinical dataset contained samples from 32 adult malaria patients, hospitalised in Hamburg, Germany (National Center for Biotechnology Information [NCBI] BioProject ID: PRJNA679547). Fifteen were malaria-naïve and 17 were previously exposed to malaria. Eight of the malaria-naïve patients went on to develop severe malaria and 24 had non-severe malaria (**Wichers et al., 2021**).

**Table 2.** Statistics for the different approaches used to assemble the *var* transcripts.
*Var* assembly approaches were applied to malaria patient ex vivo samples (n=32) from *Wichers et al., 2021*, and statistics determined. Given are the total number of assembled *var* transcripts longer than 500 nt containing at least one significantly annotated *var* domain, the maximum length of the longest assembled *var* transcript in nucleotides and the N50 value, respectively. The N50 is defined as the sequence length of the shortest *var* contig, with all *var* contigs greater than or equal to this length together accounting for 50% of the total length of concatenated *var* transcript assemblies. Misassemblies represent the number of misassemblies for each approach.

| | Number of contigs ≥500 nt | Maximum length (nt) | Average contig length (nt) | N50 | Number of misassemblies |
|---|---|---|---|---|---|
| Original approach | 6441 | 10,412 | 1621 | 2302 | 336 |
| Domain approach | 4691 | 5003 | 954 | 1088 | NA* |
| Whole transcript approach | 3011 | 12,586 | 2771 | 5381 | 2 |

*Number of misassemblies were not determined for the domain approach due to its poor performance in other metrics.

Our (i) new whole transcript approach, (ii) domain assembly approach, and (iii) modified version of the original approach (see Materials and methods) were first applied to a *P. falciparum* 3D7 time course RNA-sequencing dataset to benchmark their performance (*Wichers et al., 2019*; *Figure 2—figure supplement 1*). The whole transcript approach performed best, achieving near perfect alignment scores for the dominantly expressed *var* gene (*Figure 2—figure supplement 1a*). The domain and the original approach produced shorter contigs and required more contigs to assemble the *var* transcripts at the 8 and 16 hr post-invasion time points, when *var* gene expression is maximal (*Figure 2—figure supplement 1c, f, g, and h*). However, we found high accuracies (>0.95) across all approaches, meaning the sequences we assembled were correct (*Figure 2—figure supplement 1b*). The whole transcript approach also performed the best when assembling the lower expressed *var* genes (*Figure 2—figure supplement 1e*) and produced the fewest *var* chimeras compared to the original approach on *P. falciparum* 3D7. Fourteen misassemblies were observed with the whole transcript approach compared to 19 with the original approach (*Supplementary file 1*). This reduction in misassemblies was particularly apparent in the ring-stage samples.

Next, the assembled transcripts produced from the original approach of *Wichers et al., 2021*, were compared to those produced from our new whole transcript and domain assembly approaches for ex vivo samples from German travellers. Summary statistics are shown in *Table 2*. The whole transcript approach produced the fewest transcripts, but of greater length than the domain approach and the original approach (*Figure 2—figure supplement 2*). The whole transcript approach also returned the largest N50 score (more than doubling the N50 of the original approach), which means that it was the most contiguous assembly produced. Remarkably, with the new whole transcript method, we observed a significant decrease (2 vs 336) in clearly misassembled transcripts with, for example, an N-terminal domain at an internal position.

When genome sequencing is not available, concordance of different *var* profiling approaches can support the validation of an approach. Here, the same methods used in the original analysis were applied for quantifying the expression of the assembled *var* transcripts and domains. This suggests any concordance in expression estimates likely reflects concordance at the domain annotation level. The original approach and the new whole transcript approach gave similar results for domain expression in each sample with greater correlation in results observed between the highly expressed domains (*Figure 2—figure supplement 3*). As expected, comparable results were also seen for the differentially expressed transcripts identified in the original analysis between the naïve vs pre-exposed and severe vs non-severe comparisons, respectively (*Figure 2—figure supplement 4*).

Overall, the new whole transcript approach performed the best on the laboratory 3D7 dataset (ENA: PRJEB31535) (*Wichers et al., 2019*), had the greatest N50, the longest *var* transcripts, and produced concordant results with the original analysis on the clinical ex vivo samples (NCBI:

**Table 3.** Summary of the clinical dataset used to analyse the impact of parasite culturing on gene expression.

RNA-sequencing was performed on 32 malaria-infected German traveller samples (*Wichers et al., 2021*). The 32 ex vivo samples were used to compare the performance of the *var* assembly approaches. Parasites from 13 of these ex vivo samples underwent one cycle of in vitro replication, 10 parasite samples were also subjected to a second cycle of replication in vitro, and a single parasite isolate was also analysed after a third cycle of replication. For the ex vivo vs short-term in vitro cultivation analysis only paired samples were used. The number of assembled *var* contigs represents results per sample using the whole transcript approach, and shows either the number of assembled *var* contigs significantly annotated as *var* gene and ≥500 nt in length, or the number of assembled *var* transcripts identified with a length ≥ 1500 nt and containing at least three significantly annotated *var* domains*.

| | | Generation | | | |
|---|---|---|---|---|---|
| | | Ex vivo (n=32) | 1 (n=13) | 2 (n=10) | 3 (n=1) |
| Malaria exposure (n) | Naïve | 15 | 6 | 4 | 1 |
| | Previously exposed | 17 | 7 | 6 | 0 |
| Malaria severity (n) | Severe | 8 | 3 | 1 | 0 |
| | Non-severe | 24 | 10 | 9 | 1 |
| Number of MSP1 genotypes (number of samples) | 1 | 22 | 9 | 7 | 1 |
| | 2 | 4 | 0 | 0 | 0 |
| | 3 | 5 | 3 | 0 | 0 |
| | 4 | 1 | 1 | 0 | 0 |
| Number of *P. falciparum* PE* reads (non-*var*) (median, IQR) (million of reads) | | 34.6 (27.0–36.5) | 17.1 (12.9–18.0) | 17.2 (12.9–19.1) | 15.1 |
| Number of non-core *P. falciparum* PE* reads (median, IQR) (million of reads) | | 5.05 (3.62–6.60) | 1.16 (1.07–1.40) | 1.29 (1.04–1.58) | 0.91 |
| Number of assembled *var* contigs in a sample (≥500 nt) (whole transcript approach) (median, IQR) | | 53 (44–84) | 61 (38–76) | 71.5 (48.25–79.5) | 75 |
| Number of assembled *var* contigs in a sample (≥1500 nt and 3 sig. domain annotations) (whole transcript approach) (median, IQR) | | 20 (7–31) | 15.5 (10–26) | 15 (10.25–23.75) | 18 |

*PE, paired-end reads.

PRJNA679547) (*Wichers et al., 2021*). Therefore, it was selected for all subsequent analyses unless specified otherwise.

## Establishing characterisation of *var* transcripts from ex vivo and in vitro samples

Of the 32 clinical isolates of *P. falciparum* from the German traveller dataset, 13 underwent one replication cycle of in vitro culture, 10 of these underwent a second generation and one underwent a third generation (*Table 3*). Most (9/13, 69%) isolates entering culture had a single MSP1 genotype, indicative of monoclonal infections. All samples were sequenced with a high read depth, although the ex vivo samples had a greater read depth than the in vitro samples (*Table 3*). *Figure 3* shows a summary of the analysis performed.

To account for differences in parasite developmental stage within each sample, which are known to impact gene expression levels (*Bozdech et al., 2003*), the proportions of life cycle stages were estimated using the mixture model approach of the original analysis (*Tonkin-Hill et al., 2018*; *Wichers et al., 2021*). As a complementary approach, single-cell differential composition analysis (SCDC) with the Malaria Cell Atlas as a reference was also used to determine parasite age (*Dong et al., 2021*;

*Howick et al., 2019*). SCDC and the mixture model approaches produced concordant estimates that most parasites were at ring stage in all ex vivo and in vitro samples (*Figure 3—figure supplement 1a and b*). Whilst there was no significant difference in ring-stage proportions across the generations, we observed a slight increase in parasite age in the cultured samples. Overall, there were more rings and early trophozoites in the ex vivo samples compared to the cultured parasite samples and an increase of late trophozoite, schizont, and gametocyte proportions during the culturing process (*Figure 3—figure supplement 1c*). The estimates produced from the mixture model approach showed high concordance with those observed by counting Giemsa-stained blood smears (*Figure 3—figure supplement 1d*). Due to the potential confounding effect of differences in stage distribution on gene expression, we adjusted for developmental stage determined by the mixture model in all subsequent analyses.

Our new approach was applied to RNA-sequencing samples of ex vivo and short-term in vitro cultured parasites from German travellers (*Wichers et al., 2021*). *Supplementary file 2* shows the assembled *var* transcripts on a per sample basis. Interestingly, we observed SSPACE did not provide improvement in terms of extending *var* assembled contigs in 9/37 samples. We observed a significant increase in the number of assembled *var* transcripts in generation 2 parasites compared to paired generation 1 parasites ($p_{adj}$ = 0.04, paired Wilcoxon test). We observed no significant differences in the length of the assembled *var* transcripts across the generations. Three different filtering approaches were applied in comparison to maximise the likelihood that correct assemblies were taken forward for further analysis and to avoid the overinterpretation of lowly expressed partial *var* transcripts (*Supplementary file 3*). Filtering for *var* transcripts at least 1500 nt long and containing at least three significantly annotated *var* domains was the least restrictive, while the other approaches required the presence of a DBLα domain within the transcript. All three filtering approaches generated the same maximum length *var* transcript and similar N50 values. This suggests minimal differences in the three filtering approaches, whilst highlighting the importance of filtering assembled *var* transcripts.

In the original approach of *Wichers et al., 2021*, the non-core reads of each sample used for *var* assembly were mapped against a pooled reference of assembled *var* transcripts from all samples, as a preliminary step towards differential *var* transcript expression analysis. This approach returned a small number of *var* transcripts which were expressed across multiple patient samples (*Figure 3—figure supplement 2a*). As genome sequencing was not available, it was not possible to know whether there was truly overlap in *var* genomic repertoires of the different patient samples, but substantial overlap was not expected. Stricter mapping approaches (e.g. excluding transcripts shorter than 1500 nt) changed the resulting *var* expression profiles and produced more realistic scenarios where similar *var* expression profiles were generated across paired samples, whilst there was decreasing overlap across different patient samples (*Figure 3—figure supplement 2b and c*). Given this limitation, we used the paired samples to analyse *var* gene expression at an individual subject level, where we confirmed the MSP1 genotypes and alleles were still present after short-term in vitro cultivation. The per patient approach showed consistent expression of *var* transcripts within samples from each patient but no overlap of *var* expression profiles across different patients (*Figure 3—figure supplement 2d*). Taken together, the per patient approach was better suited for assessing *var* transcriptional changes in longitudinal samples. However, it has been hypothesised that more conserved *var* genes in field isolates increase parasite fitness during chronic infections, necessitating the need to correctly identify them (*Dimonte et al., 2020*, *Otto et al., 2019*). Accordingly, further work is needed to optimise the pooled sample approach to identify truly conserved *var* transcripts across different parasite isolates in cross-sectional studies.

## Longitudinal analysis of *var* transcriptome from ex vivo to in vitro samples

To assess the changes in the *var* transcriptome induced by parasite culturing, we performed a series of analyses, all of which addressed different aspects: (i) changes in individual *var* gene expression pattern and *var* expression homogeneity (VEH) ('per patient analysis'), (ii) changes in the expression of *var* variants conserved between strains, (iii) changes in the expression of PfEMP1 domains, (iv) changes in expression at the *var* group level, and (v) at the overall *var* expression level. We validated our results using the DBLα-tag approach and complemented the *var*-specific analysis by also examining changes in the core transcriptome.

To investigate whether dominant *var* gene expression changes through in vitro culture, rank analysis of *var* transcript expression was performed (***Figure 4***, ***Figure 4—figure supplement 1***). In most cases a single dominant *var* transcript was detected. The dominant *var* gene did not change in most patient samples and the ranking of *var* gene expression remained similar. However, we observed a change in the dominant *var* gene being expressed through culture in isolates from 3 of 13 (23%) patients (#6, #17, and #26). Changes in the dominant *var* gene expression were also observed in the DBLα-tag data for these patients (described below). In parasites from three additional patients, #1, #7, and #14, the top expressed *var* gene remained the same, however we observed a change in the ranking of other highly expressed *var* genes in the cultured samples compared to the ex vivo sample. Interestingly, in patient #26 we observed a switch from a dominant group A *var* gene to a group B and C *var* gene. This finding was also observed in the DBLα-tag analysis (results below). A similar finding was seen in patient #7. In the ex vivo sample, the second most expressed *var* transcript was a group A transcript. However, in the cultured samples expression of this transcript was reduced and we observed an increase in the expression of group B and C *var* transcripts. A similar pattern was observed in the DBLα-tag analysis for patient #7, whereby the expression of a group A transcript was reduced during the first cycle of cultivation. Overall, the data suggest that some patient samples underwent a larger *var* transcriptional change when cultured compared to the other patient samples and that culturing parasites can lead to an unpredictable *var* transcriptional change.

In line with these results, VEH on a per patient basis showed in some patients a clear change, with the ex vivo sample diversity curve distinct from those of in vitro generation 1 and generation 2 samples (patients #1, #2, #4) (***Figure 4—figure supplement 2***). Similarly, in other patient samples, we observed a clear difference in the curves of ex vivo and generation 1 samples (patients #25 and #26, both from first-time infected severe malaria patients). Some of these samples (#1 and #26, both from first-time infected severe malaria patients) also showed changes in their dominant *var* gene expression during culture, taken together indicating much greater *var* transcriptional changes in vitro compared to the other samples.

## Expression of conserved *var* gene variants through short-term in vitro culture

Due to the relatively high level of conservation observed in *var1*, *var2csa*, and *var3*, they do not present with the same limitations as regular *var* genes. Therefore, changes in their expression through short-term culture was investigated across all samples together. We observed no significant differences in the expression of conserved *var* gene variants, *var1-IT* ($p_{adj}$ = 0.61, paired Wilcoxon test), *var1-3D7* ($p_{adj}$ = 0.93, paired Wilcoxon test), and *var2csa* ($p_{adj}$ = 0.54, paired Wilcoxon test) between paired ex vivo and generation 1 parasites, but *var2csa* was significantly differentially expressed between generation 1 and generation 2 parasites ($p_{adj}$ = 0.029, paired Wilcoxon test) (***Figure 4—figure supplement 3***). However, *var2csa* expression previously appeared to have decreased in some paired samples during the first cycle of cultivation (***Figure 4—figure supplement 3***).

## Differential expression of *var* domains from ex vivo to in vitro samples

There is overlap in PfEMP1 domain subtypes of different parasite isolates which can be associated with *var* gene groups and receptor binding phenotypes. This allows performing differential expression analysis on the level of encoded PfEMP1 domain subtypes, as done in previous studies (***Tonkin-Hill et al., 2018***; ***Wichers et al., 2021***). PCA on *var* domain expression (***Figure 5a***) showed some patients' ex vivo samples clustering away from their respective generation 1 sample (patients #1, #2, #4, #12, #17, #25), again indicating a greater *var* transcriptional change relative to the other samples during the first cycle of cultivation. However, in the pooled comparison of the generation 1 vs ex vivo of all isolates, a single domain was significantly differentially expressed, CIDRα2.5 associated with B-type PfEMP1 proteins and CD36 binding (***Figure 5b***). In the generation 2 vs ex vivo comparison, there were no domains significantly differentially expressed, however we observed large $\log_2$FC values in similar domains to those changing most in the ex vivo vs generation 1 comparison (***Figure 5c***). No differentially expressed domains were found in the generation 1 vs generation 2 comparison. These results suggest that individual changes in *var* expression are not reflected in the pooled analysis and the per patient approach is more suitable.

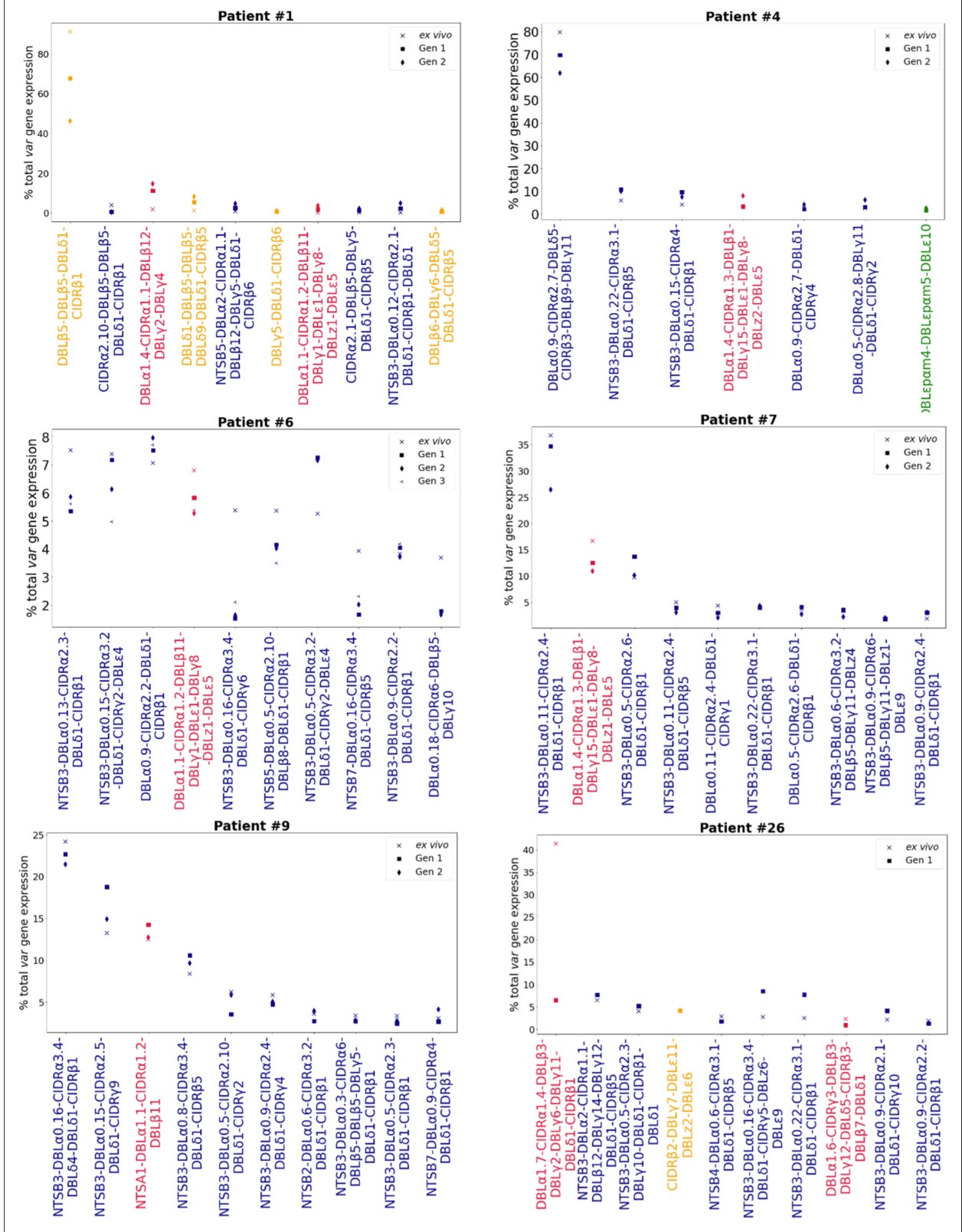

**Figure 4.** Rank *var* gene expression analysis. For each patient, the paired ex vivo (n=13) and in vitro samples (generation 1: n=13, generation 2: n=10, generation 3: n=1) were analysed. The assembled *var* transcripts with at least 1500 nt and containing three significantly annotated *var* domains across all the generations for a patient were combined into a reference, redundancy was removed using cd-hit (at sequence identity = 99%), and expression was quantified using Salmon. *Var* transcript expression was ranked. Plots show the top 10 *var* gene expression rankings for each patient and their ex vivo

*Figure 4 continued on next page*

*Figure 4 continued*

and short-term in vitro cultured parasite samples. Group A *var* transcripts (red), group B or C *var* transcripts (blue), group E *var* transcripts (green), and transcripts of unknown *var* group (orange).

The online version of this article includes the following figure supplement(s) for figure 4:

**Figure supplement 1.** Rank *var* gene expression analysis.

**Figure supplement 2.** *Var* expression homogeneity (VEH).

**Figure supplement 3.** *Var2csa* expression through short-term in vitro cultivation.

## *Var* group expression analysis

A previous study found group A *var* genes to have a rapid transcriptional decline in culture compared to group B *var* genes, however another study found a decrease in both group A and group B *var* genes in culture (*Zhang et al., 2011*; *Peters et al., 2007*). These studies were limited as the *var* type was determined by analysing the sequence diversity of DBLα domains, and by quantitative PCR (qPCR) methodology which restricts analysis to quantification of known/conserved sequences. Due to these results, the expression of group A *var* genes vs. group B and C *var* genes was investigated using a paired analysis on all the DBLα (DBLα1 vs DBLα0 and DBLα2) and NTS (NTSA vs NTSB) sequences assembled from ex vivo samples and across multiple generations in culture. A linear model was created with group A expression as the response variable, the generation and life cycle stage as independent variables, and the patient information included as a random effect. The same was performed using group B and C expression levels.

In both approaches, DBLα and NTS, we found no significant changes in total group A or group B and C *var* gene expression levels (*Figure 6*). We observed high levels of group B and C *var* gene expression compared to group A in all patients, both in the ex vivo samples and in the in vitro samples. In some patients we observed a decrease in group A *var* genes from ex vivo to generation 1 (patients #1, #2, #5, #6, #9, #12, #17, #26) (*Figure 6a*), however in all but four patients (patient #1, #2, #5, #6) the levels of group B and C *var* genes remained consistently high from ex vivo to generation 1 (*Figure 6b*). Interestingly, patients #6 and #17 also had a change in the dominant *var* gene expression through culture. Taken together with the preceding results, it appears that observed differences in *var* transcript expression occurring with transition to short-term culture are not due to modulation of recognised *var* classes, but due to differences in expression of particular *var* transcripts.

## Quantification of total *var* gene expression

We observed a trend of decreasing total *var* gene expression between generations irrespective of the assembler used in the analysis (*Figure 6—figure supplement 1*). A similar trend is seen with the LARSFADIG count, which is commonly used as a proxy for the number of different *var* genes expressed (*Otto et al., 2019*). A linear model was created (using only paired samples from ex vivo and generation 1) (*Supplementary file 4*) with proportion of total gene expression dedicated to *var* gene expression as the response variable, the generation and life cycle stage as independent variables, and the patient information included as a random effect. This model showed no significant differences between generations, suggesting that differences observed in the raw data may be a consequence of small changes in developmental stage distribution in culture.

## Validation of *var* expression profiling by DBLα-tag sequencing

Deep sequencing of RT-PCR-amplified DBLα expressed sequence tags combined with prediction of the associated transcripts and their encoded domains using the Varia tool (*Mackenzie et al., 2022*) was performed to supplement the RNA-sequencing analysis. The raw Varia output file is given in *Supplementary file 5*. Overall, we found a high agreement between the detected DBLα-tag sequences and the de novo assembled *var* transcripts. A median of 96% (IQR: 93–100%) of all unique DBLα-tag sequences detected with >10 reads were found in the RNA-sequencing approach. This is a significant improvement on the original approach (p=0.0077, paired Wilcoxon test), in which a median of 83% (IQR: 79–96%) was found (*Wichers et al., 2021*). To allow for a fair comparison of the >10 reads threshold used in the DBLα-tag approach, the upper 75th percentile of the RNA-sequencing-assembled DBLα domains were analysed. A median of 77.4% (IQR: 61–88%) of the upper

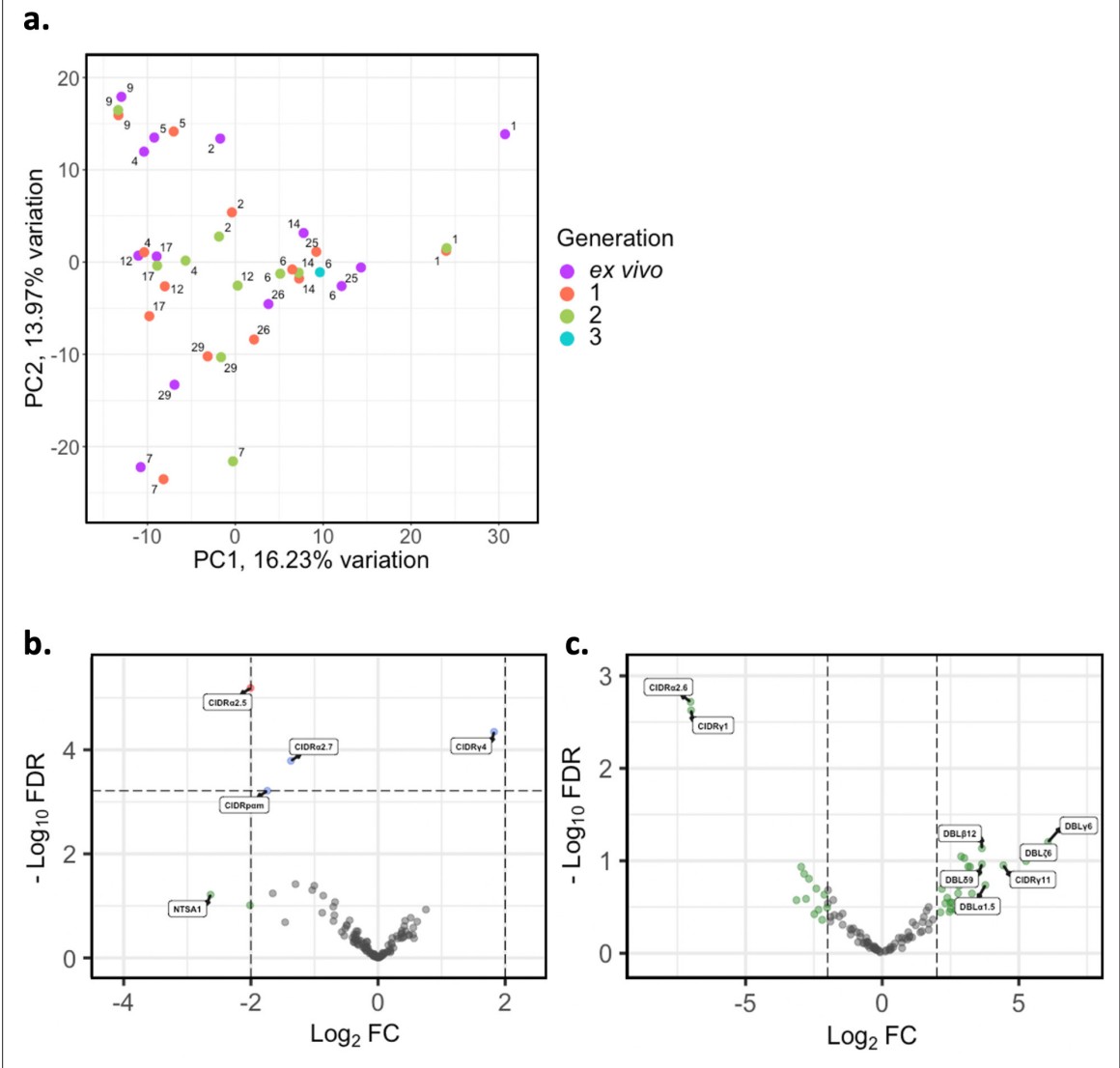

**Figure 5.** *Var* domain transcriptome analysis through short-term in vitro culture. *Var* transcripts for paired ex vivo (n=13), generation 1 (n=13), generation 2 (n=10), and generation 3 (n=1) were de novo assembled using the whole transcript approach. *Var* transcripts were filtered for those ≥ 1500 nt in length and containing at least three significantly annotated *var* domains. Transcripts were annotated using hidden Markov models (HMM) built on the *Rask et al., 2010*, dataset (*Rask et al., 2010*). When annotating the whole transcript, the most significant alignment was taken as the best annotation for each region of the assembled transcript (e-value cut-off 1e-5). Multiple annotations were allowed on the transcript if they were not overlapping, determined using cath-resolve-hits. *Var* domain expression was quantified using FeatureCounts and the domain counts aggregated. (**a**) PCA plot of $\log_2$ normalised read counts (adjusted for life cycle stage, derived from the mixture model approach). Points are coloured by their generation (ex vivo, purple; generation 1, red; generation 2, green; and generation 3, blue) and labelled by their patient identity. (**b**) Volcano plot showing extent and significance of up- or downregulation of *var* domain expression in ex vivo (n=13) compared with paired generation 1 cultured parasites (n=13) (red and blue, p<0.05 after Benjamini-Hochberg adjustment for FDR; red and green, absolute $\log_2$ fold change [$\log_2$FC] in expression ≥ 2). Domains with a $\log_2$FC ≥ 2 represent those upregulated in generation 1 parasites. Domains with a $\log_2$FC ≤ –2 represent those downregulated in generation 1 parasites. (**c**) Volcano plot showing extent and significance of up- or downregulation of *var* domain expression in ex vivo (n=10) compared with paired generation 2 cultured parasites (n=10) (green, absolute $\log_2$ fold change [$\log_2$FC] in expression ≥ 2). Domains with a $\log_2$FC ≥ 2 represent those upregulated in generation 2 parasites. Domains with a $\log_2$FC ≤ –2 represent those downregulated in generation 2 parasites. Differential expression analysis was performed using DESeq2 (adjusted for life cycle stage, derived from the mixture model approach).

75th percentile of the assembled DBLα domains were found in the DBLα-tag approach. This is a lower median percentage than the median of 81.3% (IQR: 73–98%) found in the original analysis (p=0.28, paired Wilcoxon test) and suggests the new assembly approach is better at capturing all expressed DBLα domains.

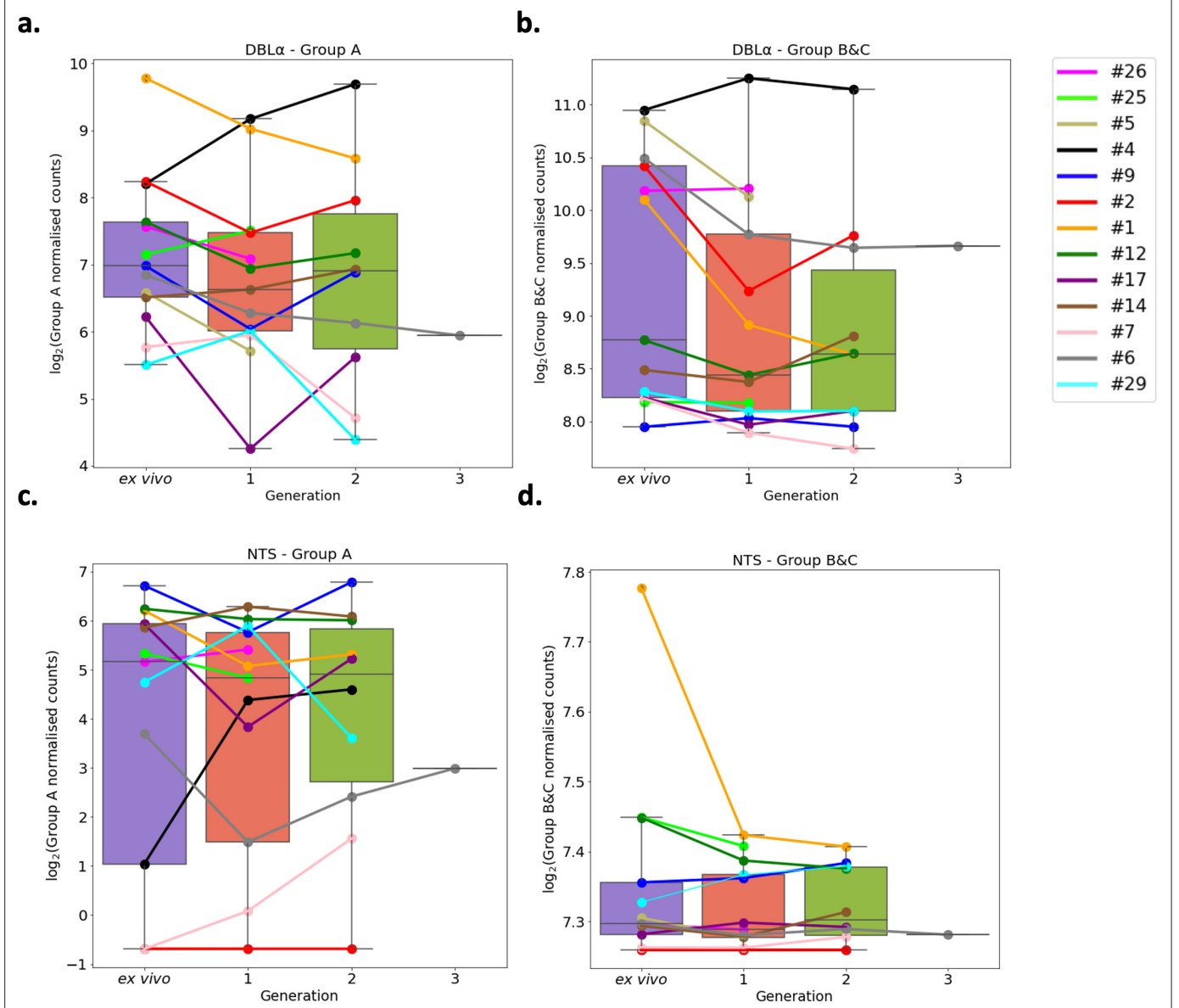

**Figure 6.** *Var* group expression analysis through short-term in vitro culture. The Duffy binding-like domain of class α (DBLα) domain sequence for each transcript was determined and for each patient a reference of all assembled DBLα domains combined. Group A *var* genes possess DBLα1 domains, some group B encode DBLα2 domains, and groups B and C encode DBLα0 domains. Domains were grouped by type and their expression summed. The relevant sample's non-core reads were mapped to this using Salmon and DBLα expression quantified. DESeq2 normalisation was performed, with patient identity and life cycle stage proportions included as covariates. A similar approach was repeated for N-terminal segment (NTS) domains. Group A *var* genes encode NTSA compared to group B and C *var* genes which encode NTSB. Boxplots show log$_2$ normalised Salmon read counts for (**a**) group A *var* gene expression through cultured generations assessed using the DBLα domain sequences, (**b**) group B and C *var* gene expression through cultured generations assessed using the DBLα domain sequences, (**c**) group A *var* gene expression through cultured generations assessed using the NTS domain sequences, and (**d**) group B and C *var* gene expression through cultured generations assessed using the NTS domain sequences. Different coloured lines connect paired patient samples through the generations: ex vivo (n=13), generation 1 (n=13), generation 2 (n=10), and generation 3 (n=1). Axis shows different scaling.

The online version of this article includes the following figure supplement(s) for figure 6:

**Figure supplement 1.** Total *var* gene expression through short-term in vitro culture.

**Figure supplement 2.** Verification of RNA-sequencing results using Duffy binding-like domain of class α (DBLα)-tag sequencing.

The new whole transcript assembly approach also had high consistency with the domain annotations predicted from Varia. Varia predicts *var* sequences and domain annotations based on short sequence tags, using a database of previously defined *var* sequences and annotations (*Mackenzie et al., 2022*). A median of 85% of the DBLα annotations and 73% of the DBLα-CIDR domain annotations, respectively, identified using the DBLα-tag approach were found in the RNA-sequencing approach. This further confirms the performance of the whole transcript approach and it was not restricted by the pooled approach of the original analysis. We also observed consistent results with the per patient analysis, in terms of changes in the dominant *var* gene expression (described above) (*Supplementary file 5*). In line with the RNA-sequencing data, the DBLα-tag approach revealed no significant differences in group A and groups B and C during short-term culture, further highlighting the agreement of both methods (*Figure 6—figure supplement 2*).

## Differential expression analysis of the core transcriptome between ex vivo and in vitro samples

Given the modest changes in *var* gene expression repertoire upon culture, we wanted to investigate the extent of any accompanying changes in the core parasite transcriptome. PCA was performed on core gene (*var, rif, stevor, surf,* and rRNA genes removed) expression, adjusted for life cycle stage. We observed distinct clustering of ex vivo, generation 1, and generation 2 samples, with patient identity having much less influence (*Figure 7a*). There was also a change from the heterogeneity between the ex vivo samples to more uniform clustering of the generation 1 samples (*Figure 7a*), suggesting that during the first cycle of cultivation the core transcriptomes of different parasite isolates become more alike.

In total, 920 core genes (19% of the core transcriptome) were found to be differentially expressed after adjusting for life cycle stages using the mixture model approach between ex vivo and generation 1 samples (*Supplementary file 6*). The majority were upregulated, indicating a substantial transcriptional change during the first cycle of in vitro cultivation (*Figure 7b*). 74 genes were found to be upregulated in generation 2 when compared to the ex vivo samples, many with $\log_2$FC greater than those in the ex vivo vs generation 1 comparison (*Figure 7c*). No genes were found to be significantly differentially expressed between generation 1 and generation 2. However, five genes had a $\log_2$FC $\geq$ 2 and were all upregulated in generation 2 compared to generation 1. Interestingly, the gene with the greatest fold change, encoding ROM3 (PF3D7_0828000), was also found to be significantly downregulated in generation 1 parasites in the ex vivo vs generation 1 analysis. The other four genes were also found to be non-significantly downregulated in generation 1 parasites in the ex vivo vs generation 1 analysis. This suggests that changes in gene expression during the first cycle of cultivation are the greatest compared to the other cycles.

The most significantly upregulated genes (in terms of fold change) in generation 1 contained several small nuclear RNAs, splicesomal RNAs, and non-coding RNAs (ncRNAs). 16 ncRNAs were found upregulated in generation 1, with several RNA-associated proteins having large fold changes ($\log_2$FC >7). Significant gene ontology (GO) terms and Kyoto encyclopedia of genes and genomes (KEGG) pathways for the core genes upregulated in generation 1 included 'entry into host', 'movement into host', and 'cytoskeletal organisation' suggesting the parasites undergo a change in invasion efficiency, which is connected to the cytoskeleton, during their first cycle of in vitro cultivation (*Figure 7—figure supplement 1*). We observed eight AP2 transcription factors upregulated in generation 1 (PF3D7_0404100/AP2-SP2, PF3D7_0604100/SIP2, PF3D7_0611200/AP2-EXP2, PF3D7_0613800, PF3D7_0802100/AP2-LT, PF3D7_1143100/AP2-O, PF3D7_1239200, PF3D7_1456000/AP2-HC) with no AP2 transcription factors found to be downregulated in generation 1. To confirm the core gene expression changes identified were not due to the increase in parasite age during culture, as indicated by upregulation of many schizont-related genes, core gene differential expression analysis was performed on paired ex vivo and generation 1 samples that contained no schizonts or gametocytes in generation 1. The same genes were identified as significantly differentially expressed with a Spearman's rank correlation of 0.99 for the $\log_2$FC correlation between this restricted sample approach and those produced using all samples (*Figure 7—figure supplement 2*).

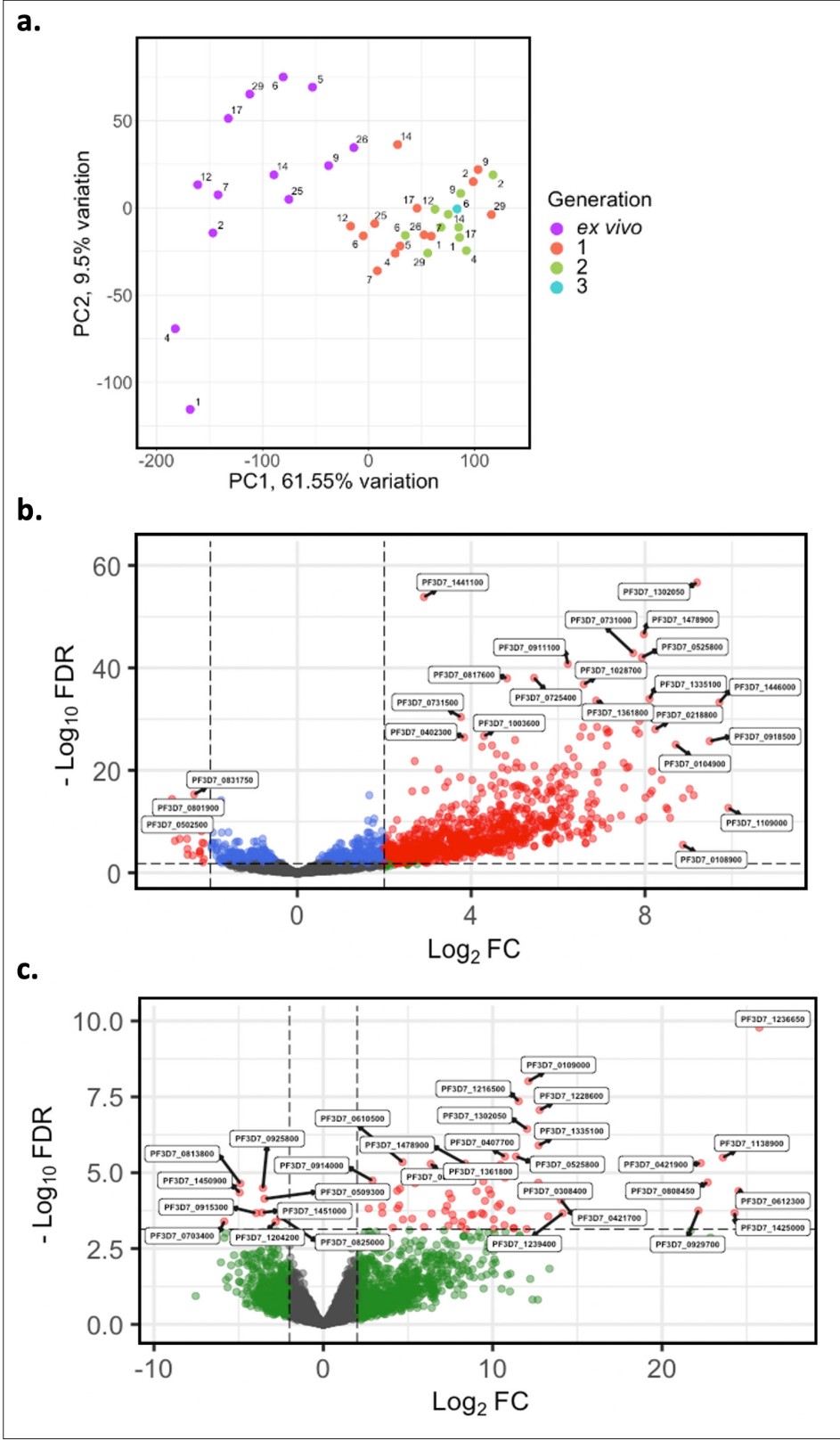

**Figure 7.** Core gene transcriptome analysis of ex vivo and short-term in vitro cultured samples. Core gene expression was assessed for paired ex vivo (n=13), generation 1 (n=13), generation 2 (n=10), and generation 3 (n=1) parasite samples. Subread align was used, as in the original analysis, to align the reads to the human genome and *P. falciparum* 3D7 genome, *with var, rif, stevor, surf,* and *rRNA* genes removed. HTSeq count was used to quantify

*Figure 7 continued on next page*

*Figure 7 continued*

gene counts. (**a**) PCA plot of log₂ normalised read counts. Points are coloured by their generation (ex vivo: purple, generation 1: red, generation 2: green, and generation 3: blue) and labelled by their patient identity. (**b**) Volcano plot showing extent and significance of up- or downregulation of core gene expression in ex vivo (n=13) compared with paired generation 1 cultured parasites (n=13) and (**c**) in ex vivo (n=10) compared with paired generation 2 cultured parasites (n=10). Dots in red and blue represent those genes with p<0.05 after Benjamini-Hochberg adjustment for FDR, red and green dots label genes with absolute log₂ fold change (log₂FC) in expression ≥ 2. Accordingly, genes with a log₂FC ≥ 2 represent those upregulated in generation 1 parasites and genes with a log₂FC ≤ –2 represent those downregulated in generation 1 parasites. Normalised read counts of the core gene analysis were adjusted for life cycle stage, derived from the mixture model approach.

The online version of this article includes the following figure supplement(s) for figure 7:

**Figure supplement 1.** Top 20 enriched pathways in the core genes found to be significantly upregulated in generation 1 parasite samples (n=13) compared to ex vivo parasite samples (n=13).

**Figure supplement 2.** Verification of core gene expression analysis excluding schizont and gametocyte stage parasite samples.

**Figure supplement 3.** Short-term in vitro cultured parasites as surrogates for assessing the in vivo core gene transcriptome.

## Cultured parasites as surrogates for assessing the in vivo core gene transcriptome

In the original analysis of ex vivo samples, hundreds of core genes were identified as significantly differentially expressed between pre-exposed and naïve malaria patients. We investigated whether these differences persisted after in vitro cultivation. We performed differential expression analysis comparing parasite isolates from naïve (n=6) vs pre-exposed (n=7) patients, first between their ex vivo samples, and then between the corresponding generation 1 samples. Interestingly, when using the ex vivo samples, we observed 206 core genes significantly upregulated in naïve patients compared to pre-exposed patients (*Figure 7—figure supplement 3a*). Conversely, we observed no differentially expressed genes in the naïve vs pre-exposed analysis of the paired generation 1 samples (*Figure 7—figure supplement 3b*). Taken together with the preceding findings, this suggests one cycle of cultivation shifts the core transcriptomes of parasites to be more alike each other, diminishing inferences about parasite biology in vivo.

A summary describing the rationale, results, and interpretation of each approach in our analysis pipeline can be found in *Table 4*.

## Discussion

Multiple lines of evidence point to PfEMP1 as a major determinant of malaria pathogenesis, but previous approaches for characterising *var* expression profiles in field samples have limited in vivo studies of PfEMP1 function, regulation, and association with clinical symptoms (*Tarr et al., 2018*; *Lee et al., 2018*; *Warimwe et al., 2013*; *Rorick et al., 2013*; *Zhang et al., 2011*; *Taylor et al., 2002*). A more recent approach, based on RNA-sequencing, overcame many of the limitations imposed by the previous primer-based methods (*Tonkin-Hill et al., 2018*; *Wichers et al., 2021*). However, depending on the expression level and sequencing depth, *var* transcripts were found to be fragmented and only a partial reconstruction of the *var* transcriptome was achieved (*Tonkin-Hill et al., 2018*; *Wichers et al., 2021*; *Andrade et al., 2020*; *Guillochon et al., 2022*; *Yamagishi et al., 2014*). The present study developed a novel approach for *var* gene assembly and quantification that overcomes many of these limitations.

Our new approach used the most geographically diverse reference of *var* gene sequences to date, which improved the identification of reads derived from *var* transcripts. This is crucial when analysing patient samples with low parasitaemia where *var* transcripts are hard to assemble due to their low abundancy (*Guillochon et al., 2022*). Our approach has wide utility due to stable performance on both laboratory-adapted and clinical samples. Concordance in the different *var* expression profiling approaches (RNA-sequencing and DBLα-tag) on ex vivo samples increased using the new approach by 13%, when compared to the original approach (96% in the whole transcript approach compared to 83% in *Wichers et al., 2021*). This suggests the new approach provides a more accurate method

**Table 4.** Summary of the different levels of analysis performed to assess the effect of short-term parasite culturing on *var* and core gene expression, their rational, method, results, and interpretation.

| Analysis level | Analysis | Rationale | Method | Results | Interpretation |
|---|---|---|---|---|---|
| *var* transcript | Per patient expression ranking | Relative quantification of *var* transcripts over consecutive generations of parasites originating from the same patient to reveal *var* gene switching events | Combine assembled *var* transcripts for each patient into a reference and quantify expression, validated with DBLα-tag analysis | 46% of the patient samples had a change in the dominant or top 3 highest expressed *var* gene | Modest changes in most samples, but unpredictable *var* gene switching during culture in some samples |
| | Per patient *var* expression homogeneity (VEH) | Determine the overall diversity of *var* gene expression (number of different variants expressed and their abundance) to assess impact of culturing on the overall *var* gene expression pattern | Comparison of diversity curves based on per patient quantification of the *var* transcriptome | 39% of ex vivo samples diversity curves distinct from in vitro samples | Some patient samples underwent a much greater *var* transcriptional change compared to others |
| | Conserved *var* variants | Assessing and comparing the expression levels of strain-transcendent *var* gene variants (*var1*, *var2csa*, *var3*) between samples | Reference of all assembled transcripts for each conserved *var* gene and quantify expression | *var2csa* expression increases in second in vitro generation | Parasites converge to *var2csa* during short-term in vitro culture |
| *var*-encoded PfEMP1 domains | Differential expression of PfEMP1 domains | Identification, quantification, and comparison of expression levels of different *var* gene-encoded PfEMP1 domains associated with different disease manifestations | Pool all assembled *var* transcripts into a reference and quantify expression of each domain | 46% of the ex vivo samples cluster away from their in vitro samples in PCA plots, distinct clustering by in vitro generation was not observed; CIDRα2.5 significantly differentially expressed between ex vivo and generation 1 | Transition to culture results in modest modulation of particular *var* domains |
| *var* group | Expression of NTS (NTSA vs NTSB) and DBLα (DBLα1 vs DBLα0+DBLα2) | Quantification and comparison of expression levels of different *var* gene groups (group A vs. groups B and C) | Create a reference of all assembled DBLα and NTS domains for each patient and quantify expression. Validated with DBLα-tag analysis | No significant changes | No preferential up- or downregulation of certain *var* groups during transition to culture |
| Global *var* expression | LARSFADIG coverage | Assessing the overall *var* gene expression level (excluding *var2csa*) | Assemble the LARSFADIG motif and map non-core reads to quantify coverage | Trend for decrease in global *var* expression during culture, but no significant changes | Subtle reduction in global *var* gene expression may reflect increase in parasite age during culture |
| Core genes | Differential gene expression (DGE) | Assessing the impact of cultivation on the parasite core gene transcriptome | Differential expression analysis of core genes (*P. falciparum* 3D7 used as reference) | 19% of the core transcriptome significantly differentially expressed between paired ex vivo and generation 1 in vitro samples; distinct clustering by parasite generation observed; upregulation of invasion and replication-related genes in vitro | Parasites core gene expression changes substantially upon entering culture |

for characterising *var* genes, especially in samples collected directly from patients. Ultimately, this will allow a deeper understanding of relationships between *var* gene expression and clinical manifestations of malaria.

Having a low number of long contigs is desirable in any de novo assembly. This reflects a continuous assembly, as opposed to a highly fragmented one where polymorphic and repeat regions could not be resolved (*Lischer and Shimizu, 2017*). An excessive number of contigs cannot be reasonably handled computationally and results from a high level of ambiguity in the assembly (*Yang et al., 2012*). We observed a greater than 50% reduction in the number of contigs produced in our new

approach, which also had a 21% increase in the maximum length of the assembled *var* transcripts, when compared to the original approach. It doubled the assembly continuity and assembled an average of 13% more of the *var* transcripts. This was particularly apparent in the N-terminal region, which has often been poorly characterised by existing approaches. The original approach failed to assemble the N-terminal region in 58% of the samples, compared to just 4% in the new approach with assembly consistently achieved with an accuracy >90%. This is important because the N-terminal region is known to contribute to the adhesion phenotypes of most PfEMP1 proteins.

The new approach allows for *var* transcript reconstruction across a range of expression levels, which is required when characterising *var* transcripts from multi-clonal infections. Assembly completeness of the lowly expressed *var* genes increased fivefold using the new approach. Biases towards certain parasite stages have been observed in non-severe and severe malaria cases, so it is valuable to assemble the *var* transcripts from different life cycle stages (*Tonkin-Hill et al., 2018*). Our new approach is not limited by parasite stage. It was able to assemble the whole *var* transcript, in a single contig, at later stages in the *P. falciparum* 3D7 intra-erythrocytic cycle, something previously unachievable. The new approach allows for a more accurate and complete picture of the *var* transcriptome. It provides new perspectives for relating *var* expression to regulation, co-expression, epigenetics, and malaria pathogenesis. It can be applied, for example, in the analysis of patient samples with different clinical outcomes and longitudinal tracking of infections in vivo. It represents a crucial improvement for quantifying the *var* transcriptome. In this work, the improved approach for *var* gene assembly and quantification was used to characterise *var* gene expression during transition from in vivo to short-term culture.

This study had substantial power through the use of paired samples. However, many *var* gene expression studies do not have longitudinal sampling. Future work should focus on identifying the best approach for analysing the *var* transcripts in cross-sectional samples. Higher-level *var* classification systems, such as the PfEMP1 predicted binding phenotype or domain cassettes, could be applied to test for over-representation of different *var* gene features in different groups of interest, because the assumption of overlapping *var* repertoires at these levels of classification would be more realistic. This was briefly explored in our analysis through *var* domain differential expression analysis, which found minimal changes in *var* domain expression through short-term culture, supporting the per patient analysis results. This could be further improved by advancing the classifications of domain subtypes. This has recently been studied using MEME to identify short nucleotide motifs that are representative of domain subtypes (*Otto et al., 2019*). Other research could investigate clustering *var* transcripts based on sequence identity and testing for clusters associated with specific malaria disease groups.

Studies have been performed investigating differences between long-term laboratory-adapted clones and clinical isolates, with hundreds of genes found to be differentially expressed (*Hoo et al., 2019*; *Tarr et al., 2018*; *Mackinnon et al., 2009*). Surprisingly, studies investigating the impact of short-term culture on parasites are extremely limited, despite it being commonly undertaken for making inferences about the in vivo transcriptome (*Vignali et al., 2011*). Using the new *var* assembly approach, we found that *var* gene expression remains relatively stable during transition to culture. However, the conserved *var2csa* had increased expression from generation 1 to generation 2. It has previously been suggested that long-term cultured parasites converge to expressing *var2csa*, but our findings suggest this begins within two cycles of cultivation (*Zhang et al., 2022*; *Mok et al., 2008*). Switching to *var2csa* has been shown to be favourable and is suggested to be the default *var* gene upon perturbation to *var*-specific heterochromatin (*Ukaegbu et al., 2015*). These studies also suggested that *var2csa* has a unique role in *var* gene switching and our results are consistent with the role of *var2csa* as the dominant 'sink node' (*Zhang et al., 2022*; *Ukaegbu et al., 2015*; *Ukaegbu et al., 2014*; *Mok et al., 2008*). A previous study suggested that in vitro cultivation of controlled human malaria infection samples resulted in dramatic changes in *var* gene expression (*Lavstsen et al., 2005*; *Peters et al., 2007*). Almost a quarter of samples in our analysis showed more pronounced and unpredictable changes. In these individuals, the dominant *var* gene being expressed changed within one cycle of cultivation. This implies that short-term culture can result in unpredictable *var* gene expression as observed previously using a semi-quantitative RT-PCR approach (*Bachmann et al., 2011*) and that one would need to confirm in vivo expression matches in vitro expression. This can be achieved using the assembly approach described here.

We observed no generalised pattern of up- or downregulation of specific *var* groups following transition to culture. This implies that there is probably not a selection event occurring during culture but may represent a loss of selection that is present in vivo. A global downregulation of certain *var* groups might only occur as a selective process over many cycles in extended culture. Determining changes in *var* group expression levels are difficult using degenerate qPCR primers bias and previous studies have found conflicting results in terms of changes of expression of *var* groups through cultivation. *Zhang et al., 2011*, found a rapid transcriptional decline of group A and group B *var* genes, however *Peters et al., 2007*, found group A *var* genes to have a high rate of downregulation, when compared to group B *var* genes. These studies differed in the stage distribution of the parasites and were limited in measuring enough variants through their use of primers. Our new approach allowed for the identification of more sequences, with 26.6% of assembled DBLα domains not found via the DBLα-tag approach. This better coverage of the expressed *var* diversity was not possible in these previous studies and may explain discrepancies observed.

Generally, there was a high consensus between all levels of *var* gene analysis and changes observed during short-term in vitro cultivation. However, the impact of short-term culture was the most apparent at the *var* transcript level and became less clear at the *var* domain, *var* type and global *var* gene expression level. This highlights the need for accurate characterisation of full-length *var* transcripts and analysis of the *var* transcriptome at different levels, both of which can be achieved with the new approach developed here.

We saw striking changes in the core gene transcriptomes between ex vivo and generation 1 parasites with 19% of the core genome being differentially expressed. A previous study showed that expression of 18% of core genes were significantly altered after ~50 cycles through culture (*Mackinnon et al., 2009*), but our data suggest that much of this change occurs early in the transition to culture. We observed that genes with functions unrelated to ring-stage parasites were among those most significantly expressed in the generation 1 vs ex vivo analysis, suggesting the culture conditions may temporarily dysregulate stage-specific expression patterns or result in the parasites undergoing a rapid adaptation response (*Andreadaki et al., 2020*; *Beeson et al., 2016*). Several AP2 transcription factors (AP2-SP2, AP2-EXP2, AP2-LT, AP2-O, and AP2-HC) were upregulated in generation 1. AP2-HC has been shown to be expressed in asexual parasites (*Carrington et al., 2021*). AP2-O is thought to be specific for the ookinete stage and AP2-SP2 plays a key role in sporozoite stage-specific gene expression (*Kaneko et al., 2015*; *Yuda et al., 2010*). Our findings are consistent with another study investigating the impact of long-term culture (*Mackinnon et al., 2009*) which also found genes like merozoite surface proteins differentially expressed, however they were downregulated in long-term cultured parasites, whereas we found them upregulated in generation 1. This suggests that short-term cultured parasites might be transcriptionally different from long-term cultured parasites, especially in their invasion capabilities, something previously unobserved. Several genes involved in the stress response of parasites were upregulated in generation 1, e.g., DnaJ proteins, serine proteases, and ATP-dependent CLP proteases (*Oakley et al., 2007*). The similarity of the core transcriptomes of the in vitro samples compared to the heterogeneity seen in the ex vivo samples could be explained by a stress response upon entry to culture. Studies investigating whether the dysregulation of stage-specific expression and the expression of stress-associated genes persist in long-term culture are required to understand whether they are important for growth in culture. Critically, the marked differences presented here suggest that the impact of short-term culture can override differences observed in both the in vivo core and *var* transcriptomes of different disease manifestations.

In summary, we present an enhanced approach for *var* transcript assembly which allows for *var* gene expression to be studied in connection to *P. falciparum's* core transcriptome through RNA-sequencing. This will be useful for expanding our understanding of *var* gene regulation and function in in vivo samples. As an example of the capabilities of the new approach, the method was used to quantify differences in gene expression upon short-term culture adaptation. This revealed that inferences from clinical isolates of *P. falciparum* put into short-term culture must be made with a degree of caution. Whilst *var* gene expression is often maintained, unpredictable switching does occur, necessitating that the similarity of in vivo and in vitro expression should be confirmed. The more extreme changes in the core transcriptome could have much bigger implications for understanding other aspects of parasite biology such as growth rates and drug susceptibility and raise a need for additional caution. Further work is needed to examine *var* and core transcriptome changes during longer-term

culture on a larger sample size. Understanding the ground truth of the *var* expression repertoire of *Plasmodium* field isolates still presents a unique challenge and this work expands the database of *var* sequences globally. The increase in long-read sequencing and the growing size of *var* gene databases containing isolates from across the globe will help overcome this issue in future studies.

## Materials and methods

**Key resources table**

| Reagent type (species) or resource | Designation | Source or reference | Identifiers | Additional information |
|---|---|---|---|---|
| Biological sample (*Plasmodium falciparum*) | Isolate #1 | Patient isolate *Wichers et al., 2021* | | Freshly isolated from a naïve, severely ill malaria patient |
| Biological sample (*Plasmodium falciparum*) | Isolate #2 | Patient isolate *Wichers et al., 2021* | | Freshly isolated from a pre-exposed, non-severely ill malaria patient |
| Biological sample (*Plasmodium falciparum*) | Isolate #3 | Patient isolate *Wichers et al., 2021* | | Freshly isolated from a naïve, non-severely ill malaria patient |
| Biological sample (*Plasmodium falciparum*) | Isolate #4 | Patient isolate *Wichers et al., 2021* | | Freshly isolated from a pre-exposed, non-severely ill malaria patient |
| Biological sample (*Plasmodium falciparum*) | Isolate #5 | Patient isolate *Wichers et al., 2021* | | Freshly isolated from a pre-exposed, non-severely ill malaria patient |
| Biological sample (*Plasmodium falciparum*) | Isolate #6 | Patient isolate *Wichers et al., 2021* | | Freshly isolated from a naïve, non-severely ill malaria patient |
| Biological sample (*Plasmodium falciparum*) | Isolate #7 | Patient isolate *Wichers et al., 2021* | | Freshly isolated from a pre-exposed, non-severely ill malaria patient |
| Biological sample (*Plasmodium falciparum*) | Isolate #9 | Patient isolate *Wichers et al., 2021* | | Freshly isolated from a naïve, non-severely ill malaria patient |
| Biological sample (*Plasmodium falciparum*) | Isolate #10 | Patient isolate *Wichers et al., 2021* | | Freshly isolated from a pre-exposed, non-severely ill malaria patient |
| Biological sample (*Plasmodium falciparum*) | Isolate #11 | Patient isolate *Wichers et al., 2021* | | Freshly isolated from a pre-exposed, non-severely ill malaria patient |
| Biological sample (*Plasmodium falciparum*) | Isolate #12 | Patient isolate *Wichers et al., 2021* | | Freshly isolated from a pre-exposed, non-severely ill malaria patient |
| Biological sample (*Plasmodium falciparum*) | Isolate #13 | Patient isolate *Wichers et al., 2021* | | Freshly isolated from a naïve, severely ill malaria patient |
| Biological sample (*Plasmodium falciparum*) | Isolate #14 | Patient isolate *Wichers et al., 2021* | | Freshly isolated from a naïve, non-severely ill malaria patient |
| Biological sample (*Plasmodium falciparum*) | Isolate #15 | Patient isolate *Wichers et al., 2021* | | Freshly isolated from a naïve, severely ill malaria patient |
| Biological sample (*Plasmodium falciparum*) | Isolate #16 | Patient isolate *Wichers et al., 2021* | | Freshly isolated from a naïve, non-severely ill malaria patient |

*Continued on next page*

*Continued*

| Reagent type (species) or resource | Designation | Source or reference | Identifiers | Additional information |
|---|---|---|---|---|
| Biological sample (*Plasmodium falciparum*) | Isolate #17 | Patient isolate *Wichers et al., 2021* | | Freshly isolated from a pre-exposed, non-severely ill malaria patient |
| Biological sample (*Plasmodium falciparum*) | Isolate #18 | Patient isolate *Wichers et al., 2021* | | Freshly isolated from a pre-exposed, non-severely ill malaria patient |
| Biological sample (*Plasmodium falciparum*) | Isolate #19 | Patient isolate *Wichers et al., 2021* | | Freshly isolated from a pre-exposed, non-severely ill malaria patient |
| Biological sample (*Plasmodium falciparum*) | Isolate #20 | Patient isolate *Wichers et al., 2021* | | Freshly isolated from a naïve, severely ill malaria patient |
| Biological sample (*Plasmodium falciparum*) | Isolate #21 | Patient isolate *Wichers et al., 2021* | | Freshly isolated from a naïve, non-severely ill malaria patient |
| Biological sample (*Plasmodium falciparum*) | Isolate #22 | Patient isolate *Wichers et al., 2021* | | Freshly isolated from a naïve, non-severely ill malaria patient |
| Biological sample (*Plasmodium falciparum*) | Isolate #23 | Patient isolate *Wichers et al., 2021* | | Freshly isolated from a naïve, severely ill malaria patient |
| Biological sample (*Plasmodium falciparum*) | Isolate #24 | Patient isolate *Wichers et al., 2021* | | Freshly isolated from a pre-exposed, non-severely ill malaria patient |
| Biological sample (*Plasmodium falciparum*) | Isolate #25 | Patient isolate *Wichers et al., 2021* | | Freshly isolated from a naïve, severely ill malaria patient |
| Biological sample (*Plasmodium falciparum*) | Isolate #26 | Patient isolate *Wichers et al., 2021* | | Freshly isolated from a naïve, severely ill malaria patient |
| Biological sample (*Plasmodium falciparum*) | Isolate #27 | Patient isolate *Wichers et al., 2021* | | Freshly isolated from a pre-exposed, non-severely ill malaria patient |
| Biological sample (*Plasmodium falciparum*) | Isolate #28 | Patient isolate *Wichers et al., 2021* | | Freshly isolated from a pre-exposed, non-severely ill malaria patient |
| Biological sample (*Plasmodium falciparum*) | Isolate #29 | Patient isolate *Wichers et al., 2021* | | Freshly isolated from a pre-exposed, non-severely ill malaria patient |
| Biological sample (*Plasmodium falciparum*) | Isolate #30 | Patient isolate *Wichers et al., 2021* | | Freshly isolated from a naïve, severely ill malaria patient |
| Biological sample (*Plasmodium falciparum*) | Isolate #31 | Patient isolate *Wichers et al., 2021* | | Freshly isolated from a pre-exposed, non-severely ill malaria patient |
| Biological sample (*Plasmodium falciparum*) | Isolate #32 | Patient isolate *Wichers et al., 2021* | | Freshly isolated from a pre-exposed, non-severely ill malaria patient |
| Biological sample (*Plasmodium falciparum*) | Isolate #33 | Patient isolate *Wichers et al., 2021* | | Freshly isolated from a pre-exposed, non-severely ill malaria patient |

*Continued*

| Reagent type (species) or resource | Designation | Source or reference | Identifiers | Additional information |
|---|---|---|---|---|
| Others | Plasmodipur filter | EuroProxima | Transia Cat. #: 8011Filter25u | Filter to remove residual granulocytes from erythrocyte pellet after Ficoll gradient centrifugation (see Materials and methods section 'Blood sampling, processing, and in vitro cultivation of *P. falciparum*') |
| Others | TRIzol | ThermoFisher | Cat. #: 15596026 | Commercial reagent |
| Commercial assay or kit | RNeasy MinElute Cleanup Kit | QIAGEN | Cat. #: 74204 | |
| Commercial assay or kit | GLOBINclear kit | ThermoFisher | Cat. #: AM1980 | |
| Commercial assay or kit | NEBNextPoly(A) mRNA Magnetic Isolation Module | NEB | Cat. #: E7490L | |
| Commercial assay or kit | NEBNext Ultra RNA Library Prep Kit for Illumina | NEB | Cat. #: E7530L | |
| Commercial assay or kit | NEBNext Multiplex Oligos for Illumina (Index Primers Set 1) | NEB | Cat. #: E7335L | |
| Commercial assay or kit | KAPA HiFi plus dNTPs | Roche | Cat. #: 7958846001 | |
| Software, algorithm | SoapDeNovo-Trans | SoapDeNovo-Trans | RRID:SCR_013268 | |
| Software, algorithm | rnaSPAdes | rnaSPAdes | RRID:SCR_016992 | |
| Software, algorithm | Oases | Oases | RRID:SCR_011896 | |
| Software, algorithm | DESeq2 | DESeq2 | RRID:SCR_015687 | |
| Software, algorithm | Salmon | Salmon | RRID:SCR_017036 | |
| Software, algorithm | CAP3 | CAP3 | RRID:SCR_007250 | |
| Software, algorithm | SSPACE | SSPACE | RRID:SCR_005056 | |
| Software, algorithm | featureCounts | featureCounts | RRID:SCR_012919 | |

## Ethics statement

The study was conducted according to the principles of the Declaration of Helsinki, 6th edition, and the International Conference on Harmonization-Good Clinical Practice (ICH-GCP) guidelines. All 32 patients were treated as inpatients or outpatients in Hamburg, Germany (outpatient clinic of the University Medical Center Hamburg-Eppendorf [UKE] at the Bernhard Nocht Institute for Tropical Medicine, UKE, Bundeswehrkrankenhaus) (*Wichers et al., 2021*). Blood samples for this analysis were collected after patients had been informed about the aims and risks of the study and had signed an informed consent form for voluntary blood collection (n=21). In the remaining cases, no intended blood samples were collected but residuals from diagnostic blood samples were used (n=11). The study was approved by the responsible ethics committee (Ethics Committee of the Hamburg Medical Association, reference numbers PV3828 and PV4539).

## Blood sampling, processing, and in vitro cultivation of *P. falciparum*

EDTA blood samples (1–30 ml) were collected from 32 adult *falciparum* malaria patients for ex vivo transcriptome profiling as reported by *Wichers et al., 2021*, hereafter termed 'the original analysis'. Blood was drawn and either immediately processed (#1, #2, #3, #4, #11, #12, #14, #17, #21, #23, #28, #29, #30, #31, #32) or stored overnight at 4°C until processing (#5, #6, #7, #9, #10, #13, #15, #16, #18, #19, #20, #22, #24, #25, #26, #27, #33). If samples were stored overnight, the ex vivo and in vitro samples were still processed at the same time (so paired samples had similar storage). Erythrocytes were isolated by Ficoll gradient centrifugation, followed by filtration through Plasmodipur filters (EuroProxima) to remove residual granulocytes. At least 400 μl of the purified erythrocytes were quickly lysed in 5 volumes of pre-warmed TRIzol (ThermoFisher Scientific) and stored at –80°C until further processing ('ex vivo samples'). When available, the remainder was then transferred to in vitro culture either without the addition of allogeneic red cells or with the addition of O+ human red cells (blood bank, UKE) for dilution according to a protocol adopted from Trager and Jensen (*Supplementary file 7*). Cultures were maintained at 37°C in an atmosphere of 1% $O_2$, 5% $CO_2$, and 94% $N_2$ using RPMI complete medium containing 10% heat-inactivated human serum (A+, Interstate Blood Bank, Inc, Memphis, TN, USA). Cultures were sampled for RNA purification at the ring stage by microscopic observation of the individual growth of parasite isolates, and harvesting was performed at the appropriate time without prior synchronisation treatment ('in vitro samples'). 13 of these ex vivo samples underwent one cycle of in vitro cultivation, 10 of these generation 1 samples underwent a second cycle of in vitro cultivation. One of these generation 2 samples underwent a third cycle of in vitro cultivation (*Table 2*). In addition, an aliquot of ex vivo erythrocytes (approximately 50–100 μl) and aliquots of in vitro cell cultures collected as indicated in *Supplementary file 8* were processed for gDNA purification and MSP1 genotyping as described elsewhere (*Wichers et al., 2021*; *Robert et al., 1996*).

## RNA purification, RNA-sequencing library preparation, and sequencing

RNA purification was performed as described in *Wichers et al., 2021*, using TRIzol in combination with the RNeasy MinElute Kit (QIAGEN) and DNase digestion (DNase I, QIAGEN). Human globin mRNA was depleted from all samples except from samples #1 and #2 using the GLOBINclear kit (ThermoFisher Scientific). The median RIN value over all ex vivo samples was 6.75 (IQR: 5.93–7.40), although this measurement has only limited significance for samples containing RNA of two species. Accordingly, the RIN value increased upon cultivation for all in vitro samples (*Supplementary file 9*). Customised library construction in accordance with *Tonkin-Hill et al., 2018*, including amplification with KAPA polymerase and HiSeq 2500 125 bp paired-end sequencing was performed by BGI Genomics Co. (Hong Kong).

## Methods for assembling *var* genes

Previously, Oases, Velvet, SoapDeNovo-Trans, or MaSuRCA have been used for *var* transcript assembly (*Wichers et al., 2021*; *Andrade et al., 2020*; *Otto et al., 2019*; *Tonkin-Hill et al., 2018*). Previous methods either did not incorporate read error correction or focussed on gene assembly, as opposed to transcript assembly (*Schulz et al., 2012*; *Zerbino and Birney, 2008*; *Xie et al., 2014*; *Zimin et al., 2013*). Read error correction is important for *var* transcript assembly due to the highly repetitive nature of the *P. falciparum* genome. Recent methods have also focussed on whole transcript assembly, as opposed to initial separate domain assembly followed by transcript assembly (*Wichers et al., 2021*; *Andrade et al., 2020*; *Otto et al., 2019*; *Tonkin-Hill et al., 2018*). The original analysis used SoapDeNovo-Trans to assemble the *var* transcripts, however it is currently not possible to run all steps in the original approach, due to certain tools being improved and updated. Therefore, SoapDeNovo-Trans (k=71) was used and termed the original approach.

Here, two novel methods for whole *var* transcript and *var* domain assembly were developed and their performance was evaluated in comparison to the original approach (*Figure 2b*). In both methods the reads were first mapped to the human g38 genome and any mapped reads were removed. Next, the unmapped reads were mapped to a modified *P. falciparum* 3D7 genome with *var* genes removed, to identify multi-mapping reads commonly present in *Plasmodium* RNA-sequencing datasets. Any mapped reads were removed. In parallel, the unmapped RNA reads from the human mapping stage were mapped against a reference of field isolate *var* exon 1 sequences and the mapped reads identified (*Otto et al., 2019*). These reads were combined with the unmapped reads from the 3D7 genome

mapping stage and duplicate reads removed. All mapping was performed using subread align as in the original analysis (*Wichers et al., 2021*). The reads identified at the end of this process are referred to as 'non-core reads'.

## Whole *var* transcript and *var* domain assembly methods

For whole *var* transcript assembly the non-core reads, for each sample separately, were assembled using rnaSPAdes (k-mer=71, read_error_correction on) (*Bushmanova et al., 2019*). Contigs were joined into larger scaffolds using SSPACE (parameters -n 31 -x 0 -k 10) (*Boetzer et al., 2011*). Transcripts <500 nt were excluded, as in the original approach. The included transcripts were annotated using hidden Markov models (HMM) (*Finn et al., 2011*) built on the *Rask et al., 2010*, dataset and used in *Tonkin-Hill et al., 2018*. When annotating the whole transcript, the most significant alignment was taken as the best annotation for each region of the assembled transcript (e-value cut-off 1e-5). Multiple annotations were allowed on the transcript if they were not overlapping, determined using cath-resolve-hits (*Lewis et al., 2019*). Scripts are available in the GitHub repository (https://github.com/ClareAndradiBrown/varAssembly, copy archived at *Brown, 2023*).

In the *var* domain assembly approach, separate domains were assembled first and then joined up to form transcripts. First, the non-core reads were mapped (nucleotide basic local alignment tool [blastn] short read option) to the domain sequences as defined in *Rask et al., 2010*. This was found to produce similar results when compared to using tblastx. An e-value threshold of 1e-30 was used for the more conserved DBLα domains and an e-value of 1e-10 for the other domains. Next, the reads mapping to the different domains were assembled separately. rnaSPAdes (read_error_correction on, k-mer=15), Oases (kmer = 15), and SoapDeNovo2 (kmer = 15) were all used to assemble the reads separately (*Bushmanova et al., 2019*; *Xie et al., 2014*; *Schulz et al., 2012*). The output of the different assemblers was combined into a per sample reference of domain sequences. Redundancy was removed in the reference using cd-hit (-n 8-c 0.99) (at sequence identity = 99%) (*Fu et al., 2012*). Cap3 was used to merge and extend the domain assemblies (*Huang and Madan, 1999*). SSPACE was used to join the domains together (parameters -n 31 -x 0 -k 10) (*Boetzer et al., 2011*). Transcript annotation was performed as in the whole transcript approach, with transcripts <500 nt removed. Significantly annotated (1e-5) transcripts were identified and selected. The most significant annotation was selected as the best annotation for each region, with multiple annotations allowed on a single transcript if the regions were not overlapping. For both methods, a *var* transcript was selected if it contained at least one significantly annotated domain (in exon 1). *Var* transcripts that encoded only the more conserved exon 2 (acidic terminal segment domain) were discarded.

## Validation on RNA-sequencing dataset from *P. falciparum* reference strain 3D7

Both new approaches and the original approach (SoapDeNovo-Trans, k=71) (*Wichers et al., 2021*; *Tonkin-Hill et al., 2018*) were run on a public RNA-sequencing dataset of the intra-erythrocytic life cycle stages of cultured *P. falciparum* 3D7 strain, sampled at 8 hr intervals up until 40 hr post infection and then at 4 hr intervals up until 48 hr post infection (ENA: PRJEB31535) (*Wichers et al., 2019*). This provided a validation of all three approaches due to the true sequence of the *var* genes being known in *P. falciparum* 3D7 strain. Therefore, we compared the assembled sequences from all three approaches to the true sequence. The first best hit (significance threshold = 1e-10) was chosen for each contig. The alignment score was used to evaluate the performance of each method. The alignment score represents √accuracy* recovery. The accuracy is the proportion of bases that are correct in the assembled transcript and the recovery reflects what proportion of the true transcript was assembled. Misassemblies were counted as transcripts that had a percentage identity <99% to their best hit (i.e. the *var* transcript is not 100% contained against the reference).

## Comparison of approaches for *var* assembly on ex vivo samples

The *var* transcripts assembled from the 32 ex vivo samples using the original approach were compared to those produced from the whole transcript and domain assembly approaches. The whole transcript approach was chosen for subsequent analysis and all assembled *var* transcripts from this approach were combined into a reference, as in the original method (*Wichers et al., 2021*).

Removal of *var* transcripts with sequence id ≥ 99% prior to mapping was not performed in the original analysis. To overcome this, *var* transcripts were removed if they had a sequence id ≥ 99% against the full complement in the whole transcript approach, using cd-hit-est (*Fu et al., 2012*). Removing redundancy in the reference of assembled *var* transcripts across all samples led to the removal of 1316 assembled contigs generated from the whole transcript approach.

This reference then represented all assembled *var* transcripts across all samples in the given analysis. The same method that was used in the original analysis was applied for quantifying the expression of the assembled *var* transcripts. The non-core reads were mapped against this reference and quantification was performed using Salmon (*Patro et al., 2017*). DESeq2 was used to perform differential expression analysis between severe versus non-severe groups and naïve versus pre-exposed groups in the original analysis (*Love et al., 2014*). Here, the same approach, as used in the original analysis, was applied to see if concordant expression estimates were obtained. As genomic sequencing was not available, this provided a confirmation of the whole transcript approach after the domain annotation step. The assembled *var* transcripts produced by the whole transcript assembly approach had their expression quantified at the transcript and domain level, as in the original method, and the results were compared to those obtained by the original method. To quantify domain expression, featureCounts was used, as in the original method with the counts for each domain aggregated (*Liao et al., 2014*). Correlation analysis between the domain's counts from the whole transcript approach and the original method was performed for each ex vivo sample. Differential expression analysis was also performed using DESeq2, as in the original analysis and the results compared (*Love et al., 2014*; *Wichers et al., 2021*).

## Estimation of parasite life cycle stage distribution in ex vivo and short-term in vitro samples

To determine the parasite life cycle stage proportions for each sample the mixture model approach of the original analysis (*Tonkin-Hill et al., 2018*; *Wichers et al., 2021*) and the SCDC approach were used (*Dong et al., 2021*; *Howick et al., 2019*). Recently, it has been determined that species-agnostic reference datasets can be used for efficient and accurate gene expression deconvolution of bulk RNA-sequencing data from any *Plasmodium* species and for correct gene expression analyses for biases caused by differences in stage composition among samples (*Tebben et al., 2022*). Therefore, the *P. berghei* single cell atlas was used as reference with restriction to 1:1 orthologs between *P. berghei* and *P. falciparum*. This reference was chosen as it contained reference transcriptomes for the gametocyte stage. To ensure consistency with the original analysis, proportions from the mixture model approach were used for all subsequent analyses (*Wichers et al., 2021*). For comparison, the proportion of different stages of the parasite life cycle in the ex vivo and in vitro samples was determined by two independent readers in Giemsa-stained thin blood smears. The same classification as the mixture model approach was used (8, 19, 30, and 42 hr post infection corresponding to ring, early trophozoite, late trophozoite, and schizont stages, respectively). Significant differences in ring-stage proportions were tested using pairwise Wilcoxon tests. For the other stages, a modified Wilcoxon rank test for zero-inflated data was used (*Wang et al., 2021*). *Var* gene expression is highly stage dependent, so any quantitative comparison between samples needs adjustment for developmental stage. The life cycle stage proportions determined from the mixture model approach were used for adjustment.

## Characterising *var* transcripts

The whole transcript approach was applied to the paired ex vivo and in vitro samples. Significant differences in the number of assembled *var* transcripts and the length of the transcripts across the generations were tested using the paired Wilcoxon test. Redundancy was removed from the assembled *var* transcripts and transcripts and domains were quantified using the approach described above. Three additional filtering steps were applied separately to this reference of assembled *var* transcripts to ensure the *var* transcripts that went on to have their expression quantified represented true *var* transcripts. The first method restricted *var* transcripts to those greater than 1500 nt containing at least three significantly annotated *var* domains, one of which had to be a DBLα domain. The second restricted *var* transcripts to those greater than 1500 nt and containing a DBLα domain. The third approach restricted *var* transcripts to those greater than 1500 nt with at least three significant *var* domain annotations.

## Per patient *var* transcript expression

A limitation of *var* transcript differential expression analysis is that it assumes that all *var* sequences have the possibility of being expressed in all samples. However, since each parasite isolate has a different set of *var* gene sequences, this assumption is not completely valid. To account for this, *var* transcript expression analysis was performed on a per patient basis. For each patient, the paired ex vivo and in vitro samples were analysed. The assembled *var* transcripts (at least 1500 nt and containing three significantly annotated *var* domains) across all the generations for a patient were combined into a reference, redundancy was removed as described above, and expression was quantified using Salmon (*Patro et al., 2017*). *Var* transcript expression was ranked, and the rankings compared across the generations.

## *Var* expression homogeneity

VEH is defined as the extent to which a small number of *var* gene sequences dominate an isolate's expression profile (*Warimwe et al., 2013*). Previously, this has been evaluated by calculating a commonly used α diversity index, the Simpson's index of diversity. Different α diversity indexes put different weights on evenness and richness. To overcome the issue of choosing one metric, α diversity curves were calculated (*Wagner et al., 2018*). *Equation 1* is the computational formula for diversity curves. D is calculated for q in the range 0–3 with a step increase of 0.1 and p in this analysis represented the proportion of *var* gene expression dedicated to *var* transcript k. q determined how much weight is given to rare vs abundant *var* transcripts. The smaller the q value, the less weight was given to the more abundant *var* transcript. VEH was investigated on a per patient basis.

$$D_{(q)} = \left(\sum_{K=1}^{K} p_k^q\right)^{\frac{1}{1-q}}$$ (1)

## Conserved *var* gene variants

To check for the differential expression of conserved *var* gene variants *var1-3D7*, *var1-IT,* and *var2csa*, all assembled transcripts significantly annotated as such were identified. For each conserved gene, Salmon normalised read counts (adjusted for life cycle stage) were summed and expression compared across the generations using a pairwise Wilcoxon rank test.

## Differential expression of *var* domains from ex vivo to in vitro samples

Domain expression was quantified using featureCounts, as described above (*Liao et al., 2014*). DESeq2 was used to test for differential domain expression, with five expected read counts in at least three patient isolates required, with life cycle stage and patient identity used as covariates. For the ex vivo versus in vitro comparisons, only ex vivo samples that had paired samples in generation 1 underwent differential expression analysis, given the extreme nature of the polymorphism seen in the *var* genes.

## *Var* group expression analysis

The type of the *var* gene is determined by multiple parameters: upstream sequence (ups), chromosomal location, direction of transcription, and domain composition. All regular *var* genes encode a DBLα domain in the N-terminus of the PfEMP1 protein (*Figure 1c*). The type of this domain correlates with previously defined *var* gene groups, with group A encoding DBLα1, groups B and C encoding DBLα0, and group B encoding a DBLα2 (chimera between DBLα0 and DBLα1) (*Figure 1c*). The DBLα domain sequence for each transcript was determined and for each patient a reference of all assembled DBLα domains combined. The relevant sample's non-core reads were mapped to this using Salmon and DBLα expression quantified (*Patro et al., 2017*). DESeq2 normalisation was performed, with patient identity and life cycle stage proportions included as covariates and differences in the amounts of *var* transcripts of group A compared with groups B and C assessed (*Love et al., 2014*). A similar approach was repeated for NTS domains. NTSA domains are found encoded in group A *var* genes and NTSB domains are found encoded in group B and C *var* genes (*Figure 1c*).

## Quantification of total *var* gene expression

The RNA-sequencing reads were blastn (with the short-blastn option on and significance = 1e-10) against the LARSFADIG nucleotide sequences (142 unique LARSFADIG sequences) to identify reads containing the LARSFADIG motifs. This approach has been described previously (*Andrade et al., 2020*). Once the reads containing the LARSFADIG motifs had been identified, they were used to assemble the LARSFADIG motif. Trinity (*Henschel et al., 2012*) and rnaSPAdes (*Bushmanova et al., 2019*) were used separately to assemble the LARSFADIG motif, and the results compared. The sequencing reads were mapped back against the assemblies using bwa mem (*Li, 2013*), parameter -k 31 -a (as in *Andrade et al., 2020*). Coverage over the LARSFADIG motif was assessed by determining the coverage over the middle of the motif (S) using Samtools depth (*Danecek et al., 2021*). These values were divided by the number of reads mapped to the *var* exon 1 database and the 3D7 genome (which had *var* genes removed) to represent the proportion of total gene expression dedicated to *var* gene expression (similar to an RPKM). The results of both approaches were compared. This method has been validated on 3D7, IT, and HB3 *Plasmodium* strains. *Var2csa* does not contain the LARSFADIG motif, hence this quantitative analysis of global *var* gene expression excluded *var2csa* (which was analysed separately). Significant differences in total *var* gene expression were tested by constructing a linear model with the proportion of gene expression dedicated to *var* gene expression as the response variable, the generation and life cycle stage as an independent variables, and the patient identity included as a random effect.

## *Var* expression profiling by DBLα-tag sequencing

DBLα-tag sequence analysis was performed as in the original analysis (*Wichers et al., 2021*), with Varia used to predict domain composition (*Mackenzie et al., 2022*). The proportion of transcripts encoding NTSA, NTSB, DBLα1, DBLα2, and DBLα0 domains were determined for each sample. These expression levels were used as an alternative approach to see whether there were changes in the *var* group expression levels through culture.

The consistency of domain annotations was also investigated between the DBLα-tag approach and the assembled transcripts. This was investigated on a per patient basis, with all the predicted annotations from the DBLα-tag approach for a given patient combined. These were compared to the annotations from all assembled transcripts for a given patient. DBLα annotations and DBLα-CIDR annotations were compared. This provided another validation of the whole transcript approach after the domain annotation step and was not dependent on performing differential expression analysis.

For comparison of both approaches (DBLα-tag sequencing and our new whole transcript approach), the same analysis was performed as in the original analysis (*Wichers et al., 2021*). All conserved variants (*var1, var2csa,* and *var3*) were removed as they were not properly amplified by the DBLα-tag approach. To identify how many assembled transcripts, specifically the DBLα region, were found in the DBLα-tag approach, we applied BLAST. As in the original analysis, a BLAST database was created from the DBLα-tag cluster results and screened for the occurrence of those assembled DBLα regions with more than 97% seq id using the 'megablast' option. This was restricted to the assembled DBLα regions that were expressed in the top 75th percentile to allow for a fair comparison, as only DBLα-tag clusters with more than 10 reads were considered. Similarly, to identify how many DBLα-tag sequences were found in the assembled transcripts, a BLAST database was created from the assembled transcripts and screened for the occurrence of the DBLα-tag sequences with more than 97% seq id using the 'megablast' option. This was performed for each sample.

## Core gene differential expression analysis

Subread align was used, as in the original analysis, to align the reads to the human genome and *P. falciparum* 3D7 genome, *with var, rif, stevor, surf,* and *rRNA* genes removed (*Liao et al., 2013*). HTSeq count was used to quantify gene counts (*Anders et al., 2015*). DESeq2 was used to test for differentially expressed genes with five read counts in at least three samples being required (*Love et al., 2014*). Parasite life cycle stages and patient identity were included as covariates. GO and KEGG

analysis was performed using ShinyGo and significant terms were defined by having a Bonferroni-corrected p-value <0.05 (*Ge et al., 2020*).

## Acknowledgements

CAB received support from the Wellcome Trust (4-year PhD programme, grant number 220123/Z/20/Z). Infrastructure support for this research was provided by the NIHR Imperial Biomedical Research Centre and Imperial College Research Computing Service, DOI: 10.14469/hpc/2232. JSWM, YDH, and AB were funded by the German Research Foundation (DFG) grants BA 5213/3-1 (project #323759012) and BA 5213/6-1 (project #433302244). TO is supported by the Wellcome Trust grant 104111/Z/14/ZR. The funders had no role in study design, data collection, and analysis, decision to publish, or preparation of the manuscript. JB acknowledges support from Wellcome (100993/Z/13/Z).

## Additional information

### Funding

| Funder | Grant reference number | Author |
| --- | --- | --- |
| Welcome Trust | 10.35802/220123 | Clare Andradi-Brown |
| German Research Foundation | 323759012 | Anna Bachmann |
| German Research Foundation | 433302244 | Anna Bachmann |
| Wellcome Trust | 10.35802/104111 | Thomas D Otto |
| Wellcome Trust | 10.35802/100993 | Jake Baum |
| Imperial College Research Computing Service | | Clare Andradi-Brown |

The funders had no role in study design, data collection and interpretation, or the decision to submit the work for publication. For the purpose of Open Access, the authors have applied a CC BY public copyright license to any Author Accepted Manuscript version arising from this submission.

### Author contributions

Clare Andradi-Brown, Conceptualization, Data curation, Formal analysis, Validation, Visualization, Methodology, Writing – original draft, Project administration, Writing – review and editing; Jan Stephan Wichers-Misterek, Yannick D Höppner, Investigation, Writing – review and editing; Heidrun von Thien, Judith AM Scholz, Helle Hansson, Emma Filtenborg Hocke, Investigation; Tim Wolf Gilberger, Jake Baum, Supervision, Writing – review and editing; Michael F Duffy, Methodology, Writing – review and editing; Thomas Lavstsen, Resources, Methodology, Writing – review and editing; Thomas D Otto, Conceptualization, Supervision, Methodology, Writing – original draft, Writing – review and editing; Aubrey J Cunnington, Conceptualization, Resources, Supervision, Funding acquisition, Writing – original draft, Project administration, Writing – review and editing; Anna Bachmann, Conceptualization, Resources, Data curation, Supervision, Funding acquisition, Validation, Investigation, Writing – original draft, Project administration, Writing – review and editing

### Author ORCIDs

Clare Andradi-Brown ⓘ https://orcid.org/0000-0003-2461-5256
Jan Stephan Wichers-Misterek ⓘ https://orcid.org/0000-0002-0599-1742
Helle Hansson ⓘ https://orcid.org/0000-0001-6484-1165
Tim Wolf Gilberger ⓘ https://orcid.org/0000-0002-7965-8272
Thomas Lavstsen ⓘ https://orcid.org/0000-0002-3044-4249
Thomas D Otto ⓘ https://orcid.org/0000-0002-1246-7404
Aubrey J Cunnington ⓘ https://orcid.org/0000-0002-1305-3529
Anna Bachmann ⓘ https://orcid.org/0000-0001-8397-7308

### Ethics

The study was conducted according to the principles of the Declaration of Helsinki, 6th edition, and the International Conference on Harmonization-Good Clinical Practice (ICH-GCP) guidelines. All 32 patients were treated as inpatients or outpatients in Hamburg, Germany (outpatient clinic of the University Medical Center Hamburg-Eppendorf (UKE) at the Bernhard Nocht Institute for Tropical Medicine, UKE, Bundeswehrkrankenhaus) (Wichers et al., 2021). Blood samples for this analysis were collected after patients had been informed about the aims and risks of the study and had signed an informed consent form for voluntary blood collection (n=21). In the remaining cases, no intended blood samples were collected but residuals from diagnostic blood samples were used (n=11). The study was approved by the responsible ethics committee (Ethics Committee of the Hamburg Medical Association, reference numbers PV3828 and PV4539).

Reviewer #1 (Public Review): https://doi.org/10.7554/eLife.87726.3.sa1
Reviewer #2 (Public Review): https://doi.org/10.7554/eLife.87726.3.sa2
Reviewer #3 (Public Review): https://doi.org/10.7554/eLife.87726.3.sa3
Author Response https://doi.org/10.7554/eLife.87726.3.sa4

## Additional files

### Supplementary files

• Supplementary file 1. Comparison of misassemblies produced by each *var* assembly approach. Table shows the number (absolute) and proportion (relative to all assembled contigs) of misassemblies produced for the dominant *var* gene (PF3D7_0712600) assembly in the *P. falciparum* 3D7 dataset (European nucleotide archive [ENA]: PRJEB31535: A public RNA-seq dataset of the intra-erythrocytic life cycle stages of cultured *P. falciparum* 3D7 strain, sampled at 8 hr intervals up until 40 hr post infection and then at 4 hr intervals up until 48 hr post infection). A misassembly was defined as a contig whose best hit was to PF3D7_0712600 and had a sequence identity <99% (i.e. was not 100% contained within the reference *var* transcript).

• Supplementary file 2. Per sample *var* assembly results using the whole transcript approach. Patient exposure and severity refer to the original patient exposure and severity that the sample originated from. # *var* transcripts before SSPACE represents the # *var* transcripts assembled before SSPACE was applied. # *var* transcripts after SSPACE represents the # *var* transcripts assembled after SSPACE was used to join contigs. # *var* significant annotated transcripts ≥ 500 nt represent the # *var* transcripts ≥ 500 nt that contained at least one significantly annotated *var* domain. # *var* significant annotated transcripts ≥ 1500 nt and three domains represent the # *var* transcripts ≥ 1500 nt that contained at least three significantly annotated *var* domains. *Var* largest transcript (nt) represents the length of the longest assembled *var* transcript in that sample in nucleotides. *Var* N50 represents the length of the shortest *var* transcript where all transcripts greater than or equal to this length when summed together represent 50% of the total *var* transcript assembly length. # *var* transcripts ≥ 5% represents the number of *var* transcripts whose expression contributed to >5% overall *var* gene expression. # assembled LARSFADIG represents the number of assembled LARSFADIG motifs (assembled using rnaSPAdes).

• Supplementary file 3. Summary statistics of the *var* transcripts after three different filtering approaches were applied to the paired ex vivo (n=13), generation 1 (n=13), generation 2 (n=10), and generation 3 (n=1) samples. The *var* transcripts were assembled using the whole transcript approach and all samples' assemblies combined into a reference. The first approach filtered for *var* transcripts that contained at least three significantly annotated domains, one of which had to be DBLα and required the transcript to be ≥ 1500 nt in length (3 domains, ≥ 1500 nt and DBLα). The second approach filtered for *var* transcripts at least 1500 nt long and that contained a DBLα domain (≥1500 nt and DBLα). The third approach filtered for transcripts that contained at least three significantly annotated *var* domains and were at least 1500 nt in length (3 domains & ≥ 1500 nt). # Significantly annotated *var* transcripts represent the number of significantly annotated *var* transcripts in all samples combined. # Uniquely annotated *var* transcripts represent the number of unique *var* transcript annotations found in all samples combined. # *Var* transcripts (≥5 in at least 3 samples) represent the number of *var* transcripts after filtering for a Salmon estimated count of 5 in at least 3 samples (filtering threshold used prior to differential expression analysis). Max length of *var* transcript (nt) represents the longest transcript assembled in all samples combined. N50

represents the length of the shortest *var* transcript where all transcripts greater than or equal to this length when summed together represent 50% of the total *var* transcript assembly length. Transcripts were annotated using hidden Markov models (HMM) built on the *Rask et al., 2010*, dataset. When annotating the whole transcript, the most significant alignment was taken as the best annotation for each region of the assembled transcript (e-value cut-off 1e-5). Multiple annotations were allowed on the transcript if they were not overlapping, determined using cath-resolve-hits.

• Supplementary file 4. Output of the linear model to quantify total *var* gene expression during the transition of parasites into in vitro culture. *Var* gene expression as a proportion of total gene expression was used as a response variable, generation and life cycle stage as independent variables, and patient information as a random effect. Only paired samples of ex vivo parasites and generation 1 parasites were used.

• Supplementary file 5. Varia results output. For each patient sample, the corresponding paired cluster is labelled (e.g., #6_1st_invitro_cluster_154=cluster 90ev shows cluster 154 of patient #6, generation 1 sample corresponds to cluster 90 of patient #6 ex vivo sample). Identical cluster sequences across paired samples were defined as having % sequence ID = >99%.

• Supplementary file 6. Differentially expressed core genes in the ex vivo vs generation 1 paired analysis. Log2FoldChange represents the $\log_2$ fold change for the ex vivo and generation 1 analysis. Values > 0 represent genes upregulated in generation 1 samples and values < 0 represent genes downregulated in generation 1 samples. pvalue and padj represent the p-value and adjusted p-value (Benjamini-Hochberg) respectively.

• Supplementary file 7. In vitro culture conditions. In vitro culture either had the addition of O+ human red cells (necessary due to high parasitaemia or low sample volume obtained from patient) or were without allogenic red cells. Patient # represents the original ex vivo patient number that the sample derived from.

• Supplementary file 8. MSP1 genotyping results from ex vivo and in vitro adapted *P. falciparum* isolates.

• Supplementary file 9. Quality of the RNA samples analysed in this study. To characterise the overall RNA quality prior to library synthesis, the Bioanalyzer automated RNA electrophoresis system was used to visualise the samples and calculate the RIN values. The measurement of the RIN value for samples from mixed species (*Homo sapiens*, *P. falciparum*) is not very meaningful, as the RIN increases the higher the proportion of a single species. This can be observed by the increase in the RIN value during in vitro cultivation of the parasites, as the parasite RNA content increases over time. Of the four rRNA peaks visible in particular in the ex vivo samples, the inner peaks represent the 18S and 28S rRNA of *P. falciparum*, the outer peaks are of human origin.

• MDAR checklist

## Data availability

The authors confirm that the data supporting the findings of this study are available within the article and its supplementary materials. Additionally, the raw sequencing data are openly available in National Center for Biotechnology Information (NCBI) at BioProject ID: PRJNA679547 (https://www.ncbi.nlm.nih.gov/bioproject/?term=PRJNA679547). The already published laboratory dataset (3D7 time course RNA-sequencing data) was deposited in the European nucleotide archive (ENA): PRJEB31535 (https://www.ebi.ac.uk/ena/browser/view/PRJEB31535; *Wichers et al., 2019*) and is also available on plasmoDB. All scripts are available in the GitHub repository (https://github.com/ClareAndradiBrown/varAssembly copy archived at *Brown, 2023*).

The following dataset was generated:

| Author(s) | Year | Dataset title | Dataset URL | Database and Identifier |
|---|---|---|---|---|
| Wichers JS, Tonkin-Hill G, Thye T, Krumkamp R, Kreuels B, Strauss J, von Thien H, Scholz JA, Smedegaard Hansson H, Weisel Jensen R, Turner L, Lorenz FR, Schöllhorn A, Bruchhaus I, Tannich E, Fendel R, Otto TD, Lavstsen T, Gilberger TW, Duffy MF, Bachmann A | 2020 | Common virulence gene expression in adult first-time infected malaria patients and severe cases | https://www.ncbi.nlm.nih.gov/bioproject/?term=PRJNA679547 | NCBI BioProject, PRJNA679547 |

The following previously published dataset was used:

| Author(s) | Year | Dataset title | Dataset URL | Database and Identifier |
|---|---|---|---|---|
| Wichers JS, Scholz JAM, Strauss J, Witt S, Lill A, Ehnold LI, Neupert N, Liffner B, Lühken R, Petter M, Lorenzen S, Wilson DW, Löw C, Lavazec C, Bruchhaus I, Tannich E, Gilberger TW, Bachmann A | 2019 | RNA-seq of the malaria parasite *Plasmodium falciparum* 3D7 during its intraerythrocytic development cycle | https://www.ebi.ac.uk/ena/browser/view/PRJEB31535 | European nucleotide archive (ENA), PRJEB31535 |

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
