## [Editor Report · eLife assessment]

Focusing mainly on var genes, the investigators performed comprehensive computational analyses of gene expression in malaria parasites isolated from patients and assessed changes that occur as these parasites adapt to in vitro culture conditions. The study provides an improved computational pipeline for monitoring var gene expression, and importantly, the study documents changes in expression of the core genome and thus provides **important** insights into metabolic adaptations that parasites undergo while transitioning to culture conditions. The findings are **important** for their technical advances that are more rigorous than the current state-of-the-art. The **solid** data analyses, broadly support the claims with only minor weaknesses, tell us to be cautious when interpreting results obtained only from cultured parasites.

---

## [Referee Report · Reviewer #1 (Public Review)]

The authors took advantage of a large dataset of transcriptomic information obtained from parasites recovered from 35 patients. In addition, parasites from 13 of these patients were reared for 1 generation in vivo, 10 for 2 generations, and 1 for a third generation. This provided the authors with a remarkable resource for monitoring how parasites initially adapt to the environmental change of being grown in culture. They focused initially on var gene expression due to the importance of this gene family for parasite virulence, then subsequently assessed changes in the entire transcriptome. Their goal was to develop a more accurate and informative computational pipeline for assessing var gene expression and secondly, to document the adaptation process at the whole transcriptome level.

Overall, the authors were largely successful in their aims. They provide convincing evidence that their new computational pipeline is better able to assemble var transcripts and assess the structure of the encoded PfEMP1s. They can also assess var gene switching as a tool for examining antigenic variation. They also documented potentially important changes in the overall transcriptome that will be important for researchers who employ ex vivo samples for assessing things like drug sensitivity profiles or metabolic states. These are likely to be important tools and insights for researchers working on field samples.

Interestingly, the conclusions about changes in var gene expression due to the transition to in vitro culture (one of the primary goals of the paper) were somewhat difficult to assess. The authors found that in most instances, var gene expression patterns changed only modestly. However, in a few cases, more substantial changes were observed. Thus, it is difficult to make firm conclusions about how one should interpret var gene expression profiles in parasites recently placed in culture. Changes in the core transcriptome however were more pronounced, justifying the authors recommendation for caution when interpreting the results of such experiments.

---

## [Referee Report · Reviewer #2 (Public Review)]

In this study, the authors describe a pipeline to sequence expressed var genes from RNA sequencing that improves on a previous one that they had developed. Importantly, they use this approach to determine how var gene expression changes with short-term culture. Their finding of shifts in the expression of particular var genes is compelling and casts some doubt on the comparability of gene expression in short-term culture versus var expression at the time of participant sampling.

Other studies have relied on short-term culture to understand var gene expression in clinical malaria studies. This study indicates the need for caution in over-interpreting findings from these studies.

We appreciate the careful attention of the authors to our comments and the edits that have been made. One additional suggestion that would be helpful to readers is to include in Table S1 the new approach described in the manuscript. This will provide the reader a direct means of comparing what the authors have done to past work.

---

## [Referee Report · Reviewer #3 (Public Review)]

This research addresses a critical challenge in malaria research, specifically how to effectively access the highly polymorphic var gene family using short-read sequence data. The authors successfully tackled this issue by introducing an optimization of their original de novo assembler, which notably more than doubled the N50 metric and greatly improved the assembly of var genes.

The most intriguing aspect of this study lies in its methodologies, particularly the longitudinal analysis of assembled var transcripts within subjects. This approach allows for the construction of an unbiased var repertoire for each individual, free from the influence of a reference genome or other samples. These sample-specific var gene repertoires are then tracked over time in culture to evaluate the reliability of using cultured samples for inferences about in vivo expression patterns. The findings from this analysis are thought-provoking. While the authors conclude that culturing parasites can lead to unpredictable transcriptional changes, they also observe that the overall ranking of each var gene remains relatively robust over time. This resilience in the var gene ranking within individuals raises intriguing questions about the mechanisms behind var gene switching and adaptation during short-term culture.

In addition to the var gene-specific analysis, the study also delves into a comparison of ex vivo samples with generation 1 and generation 2 cultured parasites across the core genome. This analysis reveals substantial shifts in expression due to culture adaptation, shedding light on broader changes in the parasite transcriptome during short-term culture.

In summary, this research contributes to our understanding of var gene expression and potentially associations with disease. It emphasizes the importance of improved assembly techniques to access var genes and underscores the challenges of using short-term cultured parasites to infer in vivo characteristics. The longitudinal analysis approach offers a fresh perspective on var gene dynamics within individuals and highlights the need for further investigations into var gene switching and adaptation during culture.

---

## [Author Response]

The following is the authors’ response to the original reviews.

**eLife assessment:**
This important study represents a comprehensive computational analysis of *Plasmodium falciparum* gene expression, with a focus on var gene expression, in parasites isolated from patients; it assesses changes that occur as the parasites adapt to short-term in vitro culture conditions. The work provides technical advances to update a previously developed computational pipeline. Although the findings of the shifts in the expression of particular var genes have theoretical or practical implications beyond a single subfield, the results are incomplete and the main claims are only partially supported.

The authors would like to thank the reviewers and editors for their insightful and constructive assessment. We particularly appreciate the statement that our work provides a technical advance of our computational pipeline given that this was one of our main aims. To address the editorial criticisms, we have rephrased and restructured the manuscript to ensure clarity of results and to support our main claims. For the same reason, we removed the var transcript differential expression analysis, as this led to confusion.

**Public Reviews:**

**Reviewer #1:**
The authors took advantage of a large dataset of transcriptomic information obtained from parasites recovered from 35 patients. In addition, parasites from 13 of these patients were reared for 1 generation in vivo, 10 for 2 generations, and 1 for a third generation. This provided the authors with aremarkable resource for monitoring how parasites initially adapt to the environmental change of being grown in culture. They focused initially on var gene expression due to the importance of this gene family for parasite virulence, then subsequently assessed changes in the entire transcriptome. Their goal was to develop a more accurate and informative computational pipeline for assessing var gene expression and secondly, to document the adaptation process at the whole transcriptome level.Overall, the authors were largely successful in their aims. They provide convincing evidence that their new computational pipeline is better able to assemble var transcripts and assess the structure of the encoded PfEMP1s. They can also assess var gene switching as a tool for examining antigenic variation. They also documented potentially important changes in the overall transcriptome that will be important for researchers who employ ex vivo samples for assessing things like drug sensitivity profiles or metabolic states. These are likely to be important tools and insights for researchers working on field samples.One concern is that the abstract highlights "Unpredictable var gene switching..." and states that "Our results cast doubt on the validity of the common practice of using short-term cultured parasites...". This seems somewhat overly pessimistic with regard to var gene expression profiling and does not reflect the data described in the paper. In contrast, the main text of the paper repeatedly refers to "modest changes in var gene expression repertoire upon culture" or "relatively small changes in var expression from ex vivo to culture", and many additional similar assessments. On balance, it seems that transition to culture conditions causes relatively minor changes in var gene expression, at least in the initial generations. The authors do highlight that a few individuals in their analysis showed more pronounced and unpredictable changes, which certainly warrants caution for future studies but should not obscure the interesting observation that var gene expression remained relatively stable during transition to culture.

Thank you for this comment. We were happy to modify the wording in the abstract to have consistency with the results presented by highlighting that modest but unpredictable var gene switching was observed while substantial changes were found in the core transcriptome. Moreover, any differences observed in core transcriptome between ex vivo samples from naïve and pre-exposed patients are diminished after one cycle of cultivation making inferences about parasite biology in vivo impossible.

Therefore, – to our opinion – the statement in the last sentence is well supported by the data presented.

Line 43–47: “Modest but unpredictable var gene switching and convergence towards var2csa were observed in culture, along with differential expression of 19% of the core transcriptome between paired ex vivo and generation 1 samples. Our results cast doubt on the validity of the common practice of using short-term cultured parasites to make inferences about in vivo phenotype and behaviour.”Nevertheless, we would like to note that this study was in a unique position to assess changes at the individual patient level as we had successive parasite generations. This comparison is not done in most cross-sectional studies and therefore these small, unpredictable changes in the var transcriptome are missed.

**Reviewer #2:**
In this study, the authors describe a pipeline to sequence expressed var genes from RNA sequencing that improves on a previous one that they had developed. Importantly, they use this approach to determine how var gene expression changes with short-term culture. Their finding of shifts in the expression of particular var genes is compelling and casts some doubt on the comparability of gene expression in short-term culture versus var expression at the time of participant sampling. The authors appear to overstate the novelty of their pipeline, which should be better situated within the context of existing pipelines described in the literature.Other studies have relied on short-term culture to understand var gene expression in clinical malaria studies. This study indicates the need for caution in over-interpreting findings from these studies.The novel method of var gene assembly described by the authors needs to be appropriately situated within the context of previous studies. They neglect to mention several recent studies that present transcript-level novel assembly of var genes from clinical samples. It is important for them to situate their work within this context and compare and contrast it accordingly. A table comparing all existing methods in terms of pros and cons would be helpful to evaluate their method.

We are grateful for this suggestion and agree that a table comparing the pros and cons of all existing methods would be helpful for the general reader and also highlight the key advantages of our new approach. A table comparing previous methods for var gene and transcript characterisation has been added to the manuscript and is referenced in the introduction (line 107) Author response table 1.

**Author response table 1. sa4table1:** Comparison of previous var assembly approaches based on DNA- and RNA-sequencing.

Study	Assembler	k-mer	Transcript or gene assembly	Validation on reference strain(s) (Yes/No)	Validation on field strain(s) (Yes/No)	Validation across different expression levels (Yes/No)	Read length (Short/Long)	Read correction (Yes/No)	Scaffolding (Yes/No)	Var transcript filter approach	Assumption	Other limitations
Duffy et al., 2016	Oases		Transcript	No	No	No	Short	Yes	Yes	Aligned to 399 var genes with BLAST (e-value< 10^–5^)		- No quantification of misassemblies - Unable to recover full length transcript assemblies
Dara et al., 2017	Sprai and Celera (no longer maintained)	71	Gene	Yes (strain NF54)	Yes (12 UM patient samples)	NA – only genome assemblies	Long and short	Method assumes combination of long and short-read sequencing will identify errors	No	>500 bp and aligned to VarDom database	Assumes a whole genome assembly is available	- Require prior filtering of human DNA - Need a combination of short-read and long-read sequencing
Tonkin-Hill et al., 2018*	SoapDeNovo-Trans	21, 31, 41, 52 & 61	Transcript	Yes (strain ITG)	No	No	Short	No	No	>500 bp and containing a sig. annotated var domain		- Unable to fully resolve N-terminus - No quantification of misassemblies
Otto et al., 2019	Masurca +post-assembly improvements	Default	Gene	Yes (clone 3D7)	Yes (15 Pf3k reference genomes)	No	Short	Yes	Yes		Whole genome dataset	
Andrade et al., 2020	Velvet	41	Transcript	No	No	No	Short	No	No	Aligned to VarDom database		- No quantification of misassemblies
Stucke et al., 2021	rnaSPAdes	Default	Transcript	No	Yes (6 UM patient samples)	No – only the most expressed var gene	Short	Yes	Unclear	>500 bp and containing a sig. annotated var domain	Information about the true var annotation is available	- Performs de novo assembly on all non-human and *P. falciparum* mapping reads - Inconsistent results in 3 samples when comparing genomic and RNA-seq results for dominant var gene

**Reviewer #3:**
This work focuses on the important problem of how to access the highly polymorphic var gene family using short-read sequence data. The approach that was most successful, and utilized for all subsequent analyses, employed a different assembler from their prior pipeline, and impressively, more than doubles the N50 metric.The authors then endeavor to utilize these improved assemblies to assess differential RNA expression of ex vivo and short-term cultured samples, and conclude that their results "cast doubt on the validity" of using short-term cultured parasites to infer in vivo characteristics. Readers should be aware that the various approaches to assess differential expression lack statistical clarity and appear to be contradictory. Unfortunately, there is no attempt to describe the rationale for the different approaches and how they might inform one another.It is unclear whether adjusting for life-cycle stage as reported is appropriate for the var-only expression models. The methods do not appear to describe what type of correction variable (continuous/categorical) was used in each model, and there is no discussion of the impact on var vs. core transcriptome results.

We agree with the reviewer that the different methods and results of the var transcriptome analysis can be difficult to reconcile. To address this, we have included a summary table with a brief description of the rationale and results of each approach in our analysis pipeline Author response table 2.

**Author response table 2. sa4table2:** Summary of the different levels of analysis performed to assess the effect of short-term parasite culturing on *var* and core gene expression, their rational, method, results, and interpretation.

Analysis level	Analysis	Rationale	Method	Results	Interpretation
var transcript	Per patient expression ranking	Relative quantification of var transcripts over consecutive generations of parasites originating from the same patient to reveal var gene switching events	Combine assembled var transcripts for each patient into a reference and quantify expression, validated with DBLα-tag analysis	46% of the patient samples had a change in the dominant or top 3 highest expressed var gene	Modest changes in most samples, but unpredictable var gene switching during culture in some samples
Per patient var expression homogeneity (VEH)	Determine the overall diversity of var gene expression (number of different variants expressed and their abundance) to assess impact of culturing on the overall var gene expression pattern	Comparison of diversity curves based on per patient quantification of the var transcriptome	39% of ex vivo samples diversity curves distinct from in vitro samples	Some patient samples underwent a much greater var transcriptional change compared to others
Conserved var variants	Assessing and comparing the expression levels of strain-transcendent var gene variants (var1, var2csa, var3) between samples	Reference of all assembled transcripts for each conserved var gene and quantify expression	var2csa expression increases in 2nd in vitro generation	Parasites converge to var2csa during short-term in vitro culture
var-encoded PfEMP1 domains	Differential expression of PfEMP1 domains	Identification, quantification and comparison of expression levels of different var gene-encoded PfEMP1 domains associated with different disease manifestations	Pool all assembled var transcripts into a reference and quantify expression of each domain	46% of the ex vivo samples cluster away from their in vitro samples in PCA plots, distinct clustering by in vitro generation was not observed; CIDRα2.5 significantly differentially expressed between ex vivo and generation 1	Transition to culture results in modest modulation of particular var domains
var group	Expression of NTS (NTSA vs NTSB) and DBLα (DBLα1 vs DBLα0+DBLα2)	Quantification and comparison of expression levels of different var gene groups (group A vs. group B and C)	Create a reference of all assembled DBLα and NTS domains for each patient and quantify expression. Validated with DBLα-tag analysis	No significant changes	No preferential up or down regulation of certain var groups during transition to culture
Global var expression	LARSFADIG coverage	Assessing the overall var gene expression level (excluding var2csa)	Assemble the LARSFADIG motif and map non-core reads to quantify coverage	Trend for decrease in global var expression during culture, but no significant changes	Subtle reduction in global var gene expression may reflect increase in parasite age during culture
Core genes	Differential gene expression (DGE)	Assessing the impact of cultivation on the parasite core gene transcriptome	Differential expression analysis of core genes (*P. falciparum* 3D7 used as reference)	19% of the core transcriptome significantly differentially expressed between paired ex vivo and generation 1 in vitro samples; distinct clustering by parasite generation observed; upregulation of invasion and replication related genes in vitro	Parasites core gene expression changes substantially upon entering culture

Additionally, the var transcript differential expression analysis was removed from the manuscript, because this study was in a unique position to perform a more focused analysis of var transcriptional changes across paired samples, meaning the per-patient approach was more suitable. This allowed for changes in the var transcriptome to be identified that would have gone unnoticed in the traditional differential expression analysis.

We thank the reviewer for his highly important comment about adjusting for life cycle stage. Var gene expression is highly stage-dependent, so any quantitative comparison between samples does need adjustment for developmental stage. All life cycle stage adjustments were done using the mixture model proportions to be consistent with the original paper, described in the results and methods sections:

Line 219–221: “Due to the potential confounding effect of differences in stage distribution on gene expression, we adjusted for developmental stage determined by the mixture model in all subsequent analyses.”Line 722–725: “Var gene expression is highly stage dependent, so any quantitative comparison between samples needs adjustment for developmental stage. The life cycle stage proportions determined from the mixture model approach were used for adjustment.“

The rank-expression analysis did not have adjustment for life cycle stage as the values were determined as a percentage contribution to the total var transcriptome. The var group level and the global var gene expression analyses were adjusted for life cycle stages, by including them as an independent variable, as described in the results and methods sections.

Var group expression:

Line 321–326: “Due to these results, the expression of group A var genes vs. group B and C var genes was investigated using a paired analysis on all the DBLα (DBLα1 vs DBLα0 and DBLα2) and NTS (NTSA vs NTSB) sequences assembled from ex vivo samples and across multiple generations in culture. A linear model was created with group A expression as the response variable, the generation and life cycle stage as independent variables and the patient information included as a random effect. The same was performed using group B and C expression levels.“Line 784–787: “DESeq2 normalisation was performed, with patient identity and life cycle stage proportions included as covariates and differences in the amounts of var transcripts of group A compared with groups B and C assessed (Love et al., 2014). A similar approach was repeated for NTS domains.”

Gobal var gene expression:

Line 342–347: “A linear model was created (using only paired samples from ex vivo and generation 1) (Supplementary file 1) with proportion of total gene expression dedicated to var gene expression as the response variable, the generation and life cycle stage as independent variables and the patient information included as a random effect. This model showed no significant differences between generations, suggesting that differences observed in the raw data may be a consequence of small changes in developmental stage distribution in culture.”Line 804–806: “Significant differences in total var gene expression were tested by constructing a linear model with the proportion of gene expression dedicated to var gene expression as the response variable, the generation and life cycle stage as an independent variables and the patient identity included as a random effect.“

The analysis of the conserved var gene expression was adjusted for life cycle stage:

Line 766–768: “For each conserved gene, Salmon normalised read counts (adjusted for life cycle stage) were summed and expression compared across the generations using a pairwise Wilcoxon rank test.”

And life cycle stage estimates were included as covariates in the design matrix for the domain differential expression analysis:

Line 771–773: “DESeq2 was used to test for differential domain expression, with five expected read counts in at least three patient isolates required, with life cycle stage and patient identity used as covariates.”

**Reviewer #1:**
1. In the legend to Figure 1, the authors cite "Deitsch and Hviid, 2004" for the classification of different var gene types. This is not the best reference for this work. Better citations would be Kraemer and Smith, Mol Micro, 2003 and Lavstsen et al, Malaria J, 2003.

We agree and have updated the legend in Figure 1 with these references, consistent with the references cited in the introduction.

1. In Figures 2 and 3, each of the boxes in the flow charts are largely filled with empty space while the text is nearly too small to read. Adjusting the size of the text would improve legibility.

We have increased the size of the text in these figures.

1. My understanding of the computational method for assessing global var gene expression indicates an initial step of identifying reads containing the amino acid sequence LARSFADIG. It is worth noting that VAR2CSA does not contain this motif. Will the pipeline therefore miss expression of this gene, and if so, how does this affect the assessment of global var gene assessment? This seems relevant given that the authors detect increased expression of var2csa during adaptation to culture.

To address this question, we have added an explanation in the methods section to better explain our analysis. Var2csa was not captured in the global var gene expression analysis, but was analyzed separately because of its unique properties (conservation, proposed role in regulating var gene switching, slightly divergent timing of expression, translational repression).

Line 802/3: “Var2csa does not contain the LARSFADIG motif, hence this quantitative analysis of global var gene expression excluded var2csa (which was analysed separately).”

1. In Figures 4 and 7, panels a and b display virtually identical PCA plots, with the exception that panel A displays more generations. Why are both panels included? There doesn't appear to be any additional information provided by panel B.

We agree and have removed Figure 7b for the core transcriptome PCA as it did not provide any new information. The var transcript differential analysis (displayed in Figure 4) has been removed from the manuscript.

1. On line 560-567, the authors state "However, the impact of short-term culture was the most apparent at the var transcript level and became less clear at higher levels." What are the high levels being referred to here?

We have replaced this sentence to make it clearer what the different levels are (global var gene expression, var domain and var type).

Line 526/7: “However, the impact of short-term culture was the most apparent at the var transcript level and became less clear at the var domain, var type and global var gene expression level.”

**Reviewer #2:**
The authors make no mention or assessment of previously published var gene assembly methods from clinical samples that focus on genomic or transcriptomic approaches. These include:
https://pubmed.ncbi.nlm.nih.gov/28351419/

https://pubmed.ncbi.nlm.nih.gov/34846163/
These methods should be compared to the method for var gene assembly outlined by the co-authors, especially as the authors say that their method "overcomes previous limitations and outperforms current methods" (128-129). The second reference above appears to be a method to measure var expression in clinical samples and so should be particularly compared to the approach outlined by the authors.

Thank you for pointing this out. We have included the second reference in the introduction of our revised manuscript, where we refer to var assembly and quantification from RNA-sequencing data. We abstained from including the first paper in this paragraph (Dara et al., 2017) as it describes a var gene assembly pipeline and not a var transcript assembly pipeline.

Line 101–105: “While approaches for var assembly and quantification based on RNA-sequencing have recently been proposed (Wichers et al., 2021; Stucke et al., 2021; Andrade et al., 2020; TonkinHill et al., 2018, Duffy et al., 2016), these still produce inadequate assembly of the biologically important N-terminal domain region, have a relatively high number of misassemblies and do not provide an adequate solution for handling the conserved var variants (Table S1).”

Additionally, we have updated the manuscript with a table (Table S1) comparing these two methods plus other previously used var transcript/gene assembly approaches (see comment to the public reviews).

But to address this particular comment in more detail, the first paper (Dara et al., 2017) is a var gene assembly pipeline and not a var transcript assembly pipeline. It is based on assembling var exon 1 from unfished whole genome assemblies of clinical samples and requires a prior step for filtering out human DNA. The authors used two different assemblers, Celera for short reads (which is no longer maintained) and Sprai for long reads (>2000bp), but found that Celera performed worse than Sprai, and subsequently used Sprai assemblies. Therefore, this method does not appear to be suitable for assembling short reads from RNA-seq.

The second paper (Stucke et al. 2021) focusses more on enriching for parasite RNA, which precedes assembly. The capture method they describe would complement downstream analysis of var transcript assembly with our pipeline. Their assembly pipeline is similar to our pipeline as they also performed de novo assembly on all *P. falciparum* mapping and non-human mapping reads and used the same assembler (but with different parameters). They clustered sequences using the same approach but at 90% sequence identity as opposed to 99% sequence identity using our approach. Then, Stucke et al. use 500nt as a cut-off as opposed to the more stringent filtering approach used in our approach. They annotated their de novo assembled transcripts with the known amino acid sequences used in their design of the capture array; our approach does not assume prior information on the var transcripts. Finally, their approach was validated only for its ability to recover the most highly expressed var transcript in 6 uncomplicated malaria samples, and they did not assess mis-assemblies in their approach.

For the methods (619–621), were erythrocytes isolated by Ficoll gradient centrifugation at the time of collection or later?

We have updated the methods section to clarify this.

Line 586–588: “Blood was drawn and either immediately processed (#1, #2, #3, #4, #11, #12, #14, #17, #21, #23, #28, #29, #30, #31, #32) or stored overnight at 4oC until processing (#5, #6, #7, #9, #10, #13, #15, #16, #18, #19, #20, #22, #24, #25, #26, #27, #33).”

Was the current pipeline and assembly method assessed for var chimeras? This should be described.

Yes, this was quantified in the Pf 3D7 dataset and also assessed in the German traveler dataset. For the 3D7 dataset it is described in the result section and Figure S1.

Line 168–174: “However, we found high accuracies (> 0.95) across all approaches, meaning the sequences we assembled were correct (Figure 2 – Figure supplement 1b). The whole transcript approach also performed the best when assembling the lower expressed var genes (Figure 2 – Figure supplement 1e) and produced the fewest var chimeras compared to the original approach on *P. falciparum* 3D7. Fourteen misassemblies were observed with the whole transcript approach compared to 19 with the original approach (Table S2). This reduction in misassemblies was particularly apparent in the ring-stage samples.” - Figure S1:

**Author response image 1. sa4fig1:** Performance of novel computational pipelines for var assembly on *Plasmodium falciparum* 3D7: The three approaches (whole transcript: blue, domain approach: orange, original approach: green) were applied to a public RNA-seq dataset (ENA: PRJEB31535) of the intra-erythrocytic life cycle stages of 3 biological replicates of cultured *P. falciparum* 3D7, sampled at 8-hour intervals up until 40hrs post infection (bpi) and then at 4-hour intervals up until 48 (Wichers al., 2019). Boxplots show the data from the 3 biological replicates for each time point in the intra-erythrocytic life cycle: (**a**) alignment scores for the dominantly expressed var gene (PF3D7_07126m), (**b**) accuracy scores for the dominantly var gene (PF3D7_0712600), (**c**) number of contigs to assemble the dominant var gene (PF3D7_0712600), (**d**) alignment scores for a middle ranking expressed vargene (PF3D7_0937800), (**e**) alignment scores for the lowest expressed var gene (PF3D7_0200100). The first best blast hit (significance threshold = le-10) was chosen for each contig. The alignment score was used to evaluate the each method. The alignment score represents √accuracy* recovery. The accuracy is the proportion of bases that are correct in the assembled transcript and the recovery reflects what proportion of the true transcript was assembled. Assembly completeness of the dominant vargene (PF3D7 071200, length = 6648nt) for the three approaches was assessed for each biological (**f**) biological replicate 1, (**g**) biological replicate 2, (**h**) biological replicate 3. Dotted lines represent the start and end of the contigs required to assemble the vargene. Red bars represent assembled sequences relative to the dominantly whole vargene sequence, where we know the true sequence (termed “reference transcript”).

For the ex vivo samples, this has been discussed in the result section and now we also added this information to Table 1.

Line 182/3: “Remarkably, with the new whole transcript method, we observed a significant decrease (2 vs 336) in clearly misassembled transcripts with, for example, an N-terminal domain at an internal position.”Table 1 (Author response table 3) :

Line 432: "the core gene transcriptome underwent a greater change relative to the var transcriptome upon transition to culture." Can this be shown statistically? It's unclear whether the difference in the sizes of the respective pools of the core genome and the var genes may account for this observation.

We found 19% of the core transcriptome to be differentially expressed. The per patient var transcript analysis revealed individually highly variable but generally rather subtle changes in the var transcriptome. The different methods for assessing this make it difficult to statistically compare these two different results.

The feasibility of this approach for field samples should be discussed in the Discussion.

In the original manuscript we reflected on this already several times in the discussion (e.g., line 465/6; line 471–475; line 555–568). We now have added another two sentences at the end of the paragraph starting in line 449 to address this point. It reads now:

Line 442–451: “Our new approach used the most geographically diverse reference of var gene sequences to date, which improved the identification of reads derived from var transcripts. This is crucial when analysing patient samples with low parasitaemia where var transcripts are hard to assemble due to their low abundancy (Guillochon et al., 2022). Our approach has wide utility due to stable performance on both laboratory-adapted and clinical samples. Concordance in the different var expression profiling approaches (RNA-sequencing and DBLα-tag) on ex vivo samples increased using the new approach by 13%, when compared to the original approach 96% in the whole transcript approach compared to 83% in Wichers et al., 2021. This suggests the new approach provides a more accurate method for characterising var genes, especially in samples collected directly from patients. Ultimately, this will allow a deeper understanding of relationships between var gene expression and clinical manifestations of malaria.”

MINORThe plural form of PfEMP1 (PfEMP1s) is inconsistently used throughout the text.

Corrected.

404-405: statistical test for significance?

Thank you for this suggestion. We have done two comparisons between the original analysis from Wichers et al., 2021 and our new whole transcript approach to test concordance of the RNAseq approaches with the DBLα-tag approach using paired Wilcoxon tests. These comparisons suggest that our new approach has significantly increased concordance with DBLα-tag data and might be better at capturing all expressed DBLα domains than the original analysis (and the DBLα-approach), although not statistically significant. We describe this now in the result section.

Line 352–361: “Overall, we found a high agreement between the detected DBLα-tag sequences and the de novo assembled var transcripts. A median of 96% (IQR: 93–100%) of all unique DBLα-tag sequences detected with >10 reads were found in the RNA-sequencing approach. This is a significant improvement on the original approach (p = 0.0077, paired Wilcoxon test), in which a median of 83% (IQR: 79–96%) was found (Wichers et al., 2021). To allow for a fair comparison of the >10 reads threshold used in the DBLα-tag approach, the upper 75th percentile of the RNA-sequencingassembled DBLα domains were analysed. A median of 77.4% (IQR: 61–88%) of the upper 75th percentile of the assembled DBLα domains were found in the DBLα-tag approach. This is a lower median percentage than the median of 81.3% (IQR: 73–98%) found in the original analysis (p = 0.28, paired Wilcoxon test) and suggests the new assembly approach is better at capturing all expressed DBLα domains.”

Figure 4: The letters for the figure panels need to be added.

The figure has been removed from the manuscript.

**Reviewer #3:**
It is difficult from Table S2 to determine how many unique var transcripts would have enough coverage to be potentially assembled from each sample. It seems unlikely that 455 distinct vars (~14 per sample) would be expressed at a detectable level for assembly. Why not DNA-sequence these samples to get the full repertoire for comparison to RNA? Why would so many distinct transcripts be yielded from fairly synchronous samples?

We know from controlled human malaria infections of malaria-naive volunteers, that most var genes present in the genomic repertoire of the parasite strain are expressed at the onset of the human blood phase (heterogenous var gene expression) (Wang et al., 2009; Bachmann et al, 2016; Wichers-Misterek et al., 2023). This pattern shifts to a more restricted, homogeneous var expression pattern in semi-immune individuals (expression of few variants) depending on the degree of immunity (Bachmann et al., 2019).

**Author response image 2. sa4fig2:** 

In this cohort, 15 first-time infections are included, which should also possess a more heterogenous var gene expression in comparison to the pre-exposed individuals, and indeed such a trend is already seen in the number of different DBLa-tag clusters found in both patient groups (see figure panel from Wichers et al. 2021: blue-first-time infections; grey–pre-exposed). Moreover, Warimwe et al. 2013 have shown that asymptomatic infections have a more homogeneous var expression in comparison to symptomatic infections. Therefore, we expect that parasites from symptomatic infections have a heterogenous var expression pattern with multiple var gene variants expressed, which we could assemble due to our high read depth and our improved var assembly pipeline for even low expressed variants.

Moreover, the distinct transcripts found in the RNA-seq approach were confirmed with the DBLα tag data. To our opinion, previous approaches may have underestimated the complexity of the var transcriptome in less immune individuals.

Mapping reads to these 455 putative transcripts and using this count matrix for differential expression analysis seems very unlikely to produce reliable results. As acknowledged on line 327, many reads will be mis-mapped, and perhaps most challenging is that most vars will not be represented in most samples. In other words, even if mapping were somehow perfect, one would expect a sparse matrix that would not be suitable for statistical comparisons between groups. This is likely why the per-patient transcript analysis doesn't appear to be consistent. I would recommend the authors remove the DE sections utilizing this approach, or add convincing evidence that the count matrix is useable.

We agree that this is a general issue of var differential expression analysis. Therefore, we have removed the var differential expression analysis from this manuscript as the per patient approach was more appropriate for the paired samples. We validated different mapping strategies (new Figure S6) and included a paragraph discussing the problem in the result section:

Line 237–255: “In the original approach of Wichers et al., 2021, the non-core reads of each sample used for var assembly were mapped against a pooled reference of assembled var transcripts from all samples, as a preliminary step towards differential var transcript expression analysis. This approach returned a small number of var transcripts which were expressed across multiple patient samples (Figure 3 – Figure supplement 2a). As genome sequencing was not available, it was not possible to know whether there was truly overlap in var genomic repertoires of the different patient samples, but substantial overlap was not expected. Stricter mapping approaches (for example, excluding transcripts shorter than 1500nt) changed the resulting var expression profiles and produced more realistic scenarios where similar var expression profiles were generated across paired samples, whilst there was decreasing overlap across different patient samples (Figure 3 – Figure supplement 2b,c). Given this limitation, we used the paired samples to analyse var gene expression at an individual subject level, where we confirmed the MSP1 genotypes and alleles were still present after short-term in vitro cultivation. The per patient approach showed consistent expression of var transcripts within samples from each patient but no overlap of var expression profiles across different patients (Figure 3 – Figure supplement 2d). Taken together, the per patient approach was better suited for assessing var transcriptional changes in longitudinal samples. It has been hypothesised that more conserved var genes in field isolates increase parasite fitness during chronic infections, necessitating the need to correctly identify them (Dimonte et al., 2020, Otto et al., 2019). Accordingly, further work is needed to optimise the pooled sample approach to identify truly conserved var transcripts across different parasite isolates in cross-sectional studies.” - Figure S6:

**Author response image 3. sa4fig3:** Var expression profiles across different mapping. Different mapping approaches Were used to quantify the Var expression profiles of each sample (ex Vivo (n=13), generation I (n=13), generation 2 (n=10) and generation 3 (n=l)). The pooled sample approach in Which all significantly assembled van transcripts (1500nt and containing3 significantly annotated var domains) across samples were combined into a reference and redundancy was removed using cd-hit (at sequence identity = 99%) (a—c). The non-core reads of each sample were mapped to this pooled reference using (a) Salmon, (b) bowtie2 filtering for uniquely mapping paired reads with MAPQ and (c) bowtie2 filtering for uniquely mapping paired reads with a MAPQ > 20. (d) The per patient approach was applied. For each patient, the paired ex vivo and in vitro samples were analysed. The assembled var transcripts (at least 1500nt and containing3 significantly annotated var domains) across all the generations for a patient were combined into a reference, redundancy was removed using cd-hit (at sequence identity: 99%), and expression was quantified using Salmon. Pie charts show the var expression profile With the relative size of each slice representing the relative percentage of total var gene expression of each var transcript. Different colours represent different assembled var transcripts with the same colour code used across (a-d).

For future cross-sectional studies a per patient analysis that attempts to group per patient assemblies on some unifying structure (e.g., domain, homology blocks, domain cassettes etc) should be performed.

Line 304. I don't understand the rationale for comparing naïve vs. prior-exposed individuals at ex-vivo and gen 1 timepoints to provide insights into how reliable cultured parasites are as a surrogate for var expression in vivo. Further, the next section (per patient) appears to confirm the significant limitation of the 'all sample analysis' approach. The conclusion on line 319 is not supported by the results reported in figures S9a and S9b, nor is the bold conclusion in the abstract about "casting doubt" on experiments utilizing culture adapted

We have removed this comparison from the manuscript due to the inconsistencies with the var per patient approach. However, the conclusion in the abstract has been rephrased to reflect the fact we observed 19% of the core transcript differentially expressed within one cycle of cultivation.

Line 372/391 (and for the other LMM descriptions). I believe you mean to say response variable, rather than explanatory variable. Explanatory variables are on the right hand side of the equation.

Thank you for spotting this inaccuracy, we changed it to “response variable” (line 324, line 343, line 805).

Line 467. Similar to line 304, why would comparisons of naïve vs. prior-exposed be informative about surrogates for in vivo studies? Without a gold-standard for what should be differentially expressed between naïve and prior-exposed in vivo, it doesn't seem prudent to interpret a drop in the number of DE genes for this comparison in generation 1 as evidence that biological signal for this comparison is lost. What if the generation 1 result is actually more reflective of the true difference in vivo, but the ex vivo samples are just noisy? How do we know? Why not just compare ex vivo vs generation 1/2 directly (as done in the first DE analysis), and then you can comment on the large number of changes as samples are less and less proximal to in vivo?

In the original paper (Wichers et al., 2021), there were differences between the core transcriptome of naïve vs previously exposed patients. However, these differences appeared to diminish in vitro, suggesting the in vivo core transcriptome is not fully maintained in vitro.

We have added a sentence explaining the reasoning behind this analysis in the results section:

Lines 414–423: “In the original analysis of ex vivo samples, hundreds of core genes were identified as significantly differentially expressed between pre-exposed and naïve malaria patients. We investigated whether these differences persisted after in vitro cultivation. We performed differential expression analysis comparing parasite isolates from naïve (n=6) vs pre-exposed (n=7) patients, first between their ex vivo samples, and then between the corresponding generation 1 samples. Interestingly, when using the ex vivo samples, we observed 206 core genes significantly upregulated in naïve patients compared to pre-exposed patients (Figure 7 – Figure supplement 3a). Conversely, we observed no differentially expressed genes in the naïve vs pre-exposed analysis of the paired generation 1 samples (Figure 7 – Figure supplement 3b). Taken together with the preceding findings, this suggests one cycle of cultivation shifts the core transcriptomes of parasites to be more alike each other, diminishing inferences about parasite biology in vivo.”

Overall, I found the many DE approaches very frustrating to interpret coherently. If not dropped in revision, the reader would benefit from a substantial effort to clarify the rationale for each approach, and how each result fits together with the other approaches and builds to a concise conclusion.

We agree that the manuscript contains many different complex layers of analysis and that it is therefore important to explain the rationale for each approach. Therefore, we now included the summary Table 3 (see comment to public review). Additionally, we have removed the var transcript differential expression due to its limitations, which we hope has already streamlined our manuscript.

**Author response table 3. sa4table3:** Statistics for the different approaches used to assemble the *var* transcripts. *Var* assembly approaches were applied to malaria patient *ex vivo* samples (n=32) from (Wichers *et al.*, 2021) and statistics determined. Given are the total number of assembled *var* transcripts longer than 500 nt containing at least one significantly annotated *var* domain, the maximum length of the longest assembled *var* transcript in nucleotides and the N50 value, respectively. The N50 is defined as the sequence length of the shortest *var* contig, with all *var* contigs greater than or equal to this length together accounting for 50% of the total length of concatenated *var* transcript assemblies. Misassemblies represents the number of misassemblies for each approach. *Number of misassemblies were not determined for the domain approach due to its poor performance in other metrics.

	Number of contigs≥ 500nts	Maximum length (nt)	Average contig length (nt)	N50	Number of misassemblies
Original approach	6,441	10,412	1,621	2,302	336
Domain approach	4,691	5,003	954	1,088	NA*
Whole transcript approach	3,011	12,586	2,771	5,381	2